# OPENMIXUP: OPEN MIXUP TOOLBOX AND BENCHMARK FOR VISUAL REPRESENTATION LEARNING

## ABSTRACT

Mixup augmentation has emerged as a widely used technique for improving the generalization ability of deep neural networks (DNNs). However, the lack of standardized implementations and benchmarks has impeded recent progress, resulting in poor reproducibility, unfair comparisons, and conflicting insights. In this paper, we introduce OpenMixup, the *first* mixup augmentation codebase and benchmark for visual representation learning. Specifically, we train 18 representative mixup baselines *from scratch* and rigorously evaluate them across 11 image datasets of varying scales and granularity, ranging from fine-grained scenarios to complex non-iconic scenes. We also open-source our modular codebase including a collection of popular vision backbones, optimization strategies, and analysis toolkits, which not only supports the benchmarking but enables broader mixup applications beyond classification, such as self-supervised learning and regression tasks. Through experiments and empirical analysis, we gain observations and insights on mixup performance-efficiency trade-offs, generalization, and optimization behaviors, and thereby identify preferred choices for different needs. To the best of our knowledge, OpenMixup has facilitated several recent studies. We believe this work can further advance reproducible mixup augmentation research and thereby lay a solid ground for future progress in the community. The source code will be publicly available.

## 1 INTRODUCTION

Data mixing, or mixup, has proven effective in enhancing the generalization ability of DNNs, with notable success in visual classification tasks. The pioneering Mixup (Zhang et al., 2018) proposes to generate mixed training examples through the convex combination of two input samples and their corresponding one-hot labels. By encouraging models to learn smoother decision boundaries, mixup effectively reduces overfitting and thus improves the overall performance. ManifoldMix (Verma et al., 2019) and PatchUp (Faramarzi et al., 2020) extend this operation to the hidden space. CutMix (Yun et al., 2019) presents an alternative approach, where an input rectangular region is randomly cut and pasted onto the target in the identical location. Subsequent works (Harris et al., 2020; ha Lee et al., 2020; Baek et al., 2021) have focused on designing more complex *hand-crafted* policies to gener-

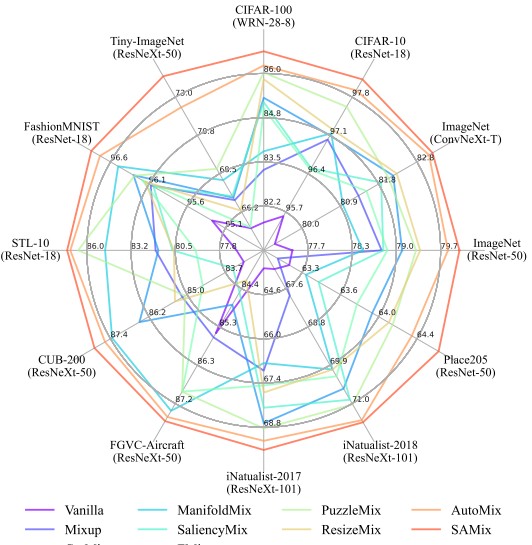

Figure 1: Radar plot of top-1 accuracy for representative mixup baselines on 11 classification datasets.

ate diverse and informative mixed samples, which can all be categorized as *static* mixing methods.

Despite efforts to incorporate saliency information into *static* mixing framework (Walawalkar et al., 2020; Uddin et al., 2020; Qin et al., 2023), they still struggle to ensure the inclusion of desired targets in the mixed samples, which may result in the issue of label mismatches. To address this problem, a new class of optimization-based methods, termed *dynamic* mixing, has been proposed, as illustrated in the second row of Figure 2. PuzzleMix (Kim et al., 2020) and Co-Mixup (Kim et al., 2021) are two notable studies that leverage optimal transport to improve offline mask determination.

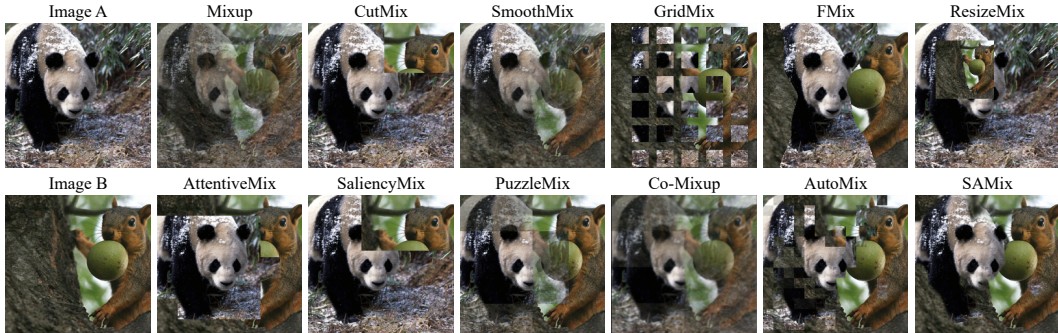

Figure 2: Visualization of mixed samples from representative *static* and *dynamic* mixup augmentation methods on ImageNet-1K. We employ a mixing ratio of $\lambda = 0.5$ for a comprehensive comparison. Note that mixed samples are more precisely in *dynamic* mixing policies than these *static* ones.

More recently, TransMix (Chen et al., 2022), TokenMix (Liu et al., 2022a), MixPro (Zhao et al., 2023), and SMMix (Chen et al., 2023) are specifically tailored for Vision Transformers (Dosovitskiy et al., 2021). The AutoMix series (Liu et al., 2022d; Qin et al., 2024) introduces a brand-new mixup learning paradigm, where mixed samples are computed by an online-optimizable generator in an end-to-end manner. These emerging *dynamic* approaches represent a promising avenue for generating semantically richer training samples that align with the underlying structure of input data.

**Why do we call for a mixup augmentation benchmark?** While *dynamic* methods have shown signs of surpassing the *static* ones, their indirect optimization process incurs significant computational overhead, which limits their efficiency and applicability. Therefore, without a systematic understanding, it is uncertain if *dynamic* mixup serves as the superior alternative in vision tasks. Moreover, a thorough and standardized evaluation of different *dynamic* methods is also missing in the community. Benchmark is exactly the way to establish such an understanding, which plays a pivotal role in driving research progress by integrating an agreed-upon set of tasks, impartial comparisons, and assessment criteria. To the best of our knowledge, however, there have been no such comprehensive benchmarks for mixup augmentation to facilitate unbiased comparisons and practical use in visual recognition.

**Why do we need an open-source mixup codebase?** Notably, most existing mixup techniques are crafted with diverse settings, tricks, and implementations, each with its own coding style. This lack of standardization not only hinders user-friendly reproduction and deployment but impedes further development, thus imposing costly trial-and-error on practitioners to determine the most appropriate mixup strategy for their specific needs in real-world applications. Hence, it is essential to develop a unified mixup visual representation learning codebase for standardized data pre-processing, mixup development, network architecture selection, model training, evaluation, and empirical analysis.

In this paper, we present OpenMixup, the *first* comprehensive benchmark for mixup augmentation in vision tasks. Unlike previous work (Naveed, 2021; Lewy & Mańdziuk, 2023), we train and evaluate 18 methods that represent the foremost strands on 11 diverse image datasets, as illustrated in Figure 1. We also open-source a standardized mixup codebase for visual representation learning, where the overall framework is built up with modular components for data pre-processing, mixup augmentation, network backbone selection, optimization, and evaluations. The codebase not only powers our benchmarking but supports broader relatively under-explored mixup applications beyond classification, such as semi-supervised learning (Berthelot et al., 2019), self-supervised learning (Kalantidis et al., 2020; Shen et al., 2022), and dense prediction tasks (He et al., 2017; Bochkovskiy et al., 2020).

Furthermore, insightful observations are obtained by incorporating multiple evaluation metrics and analysis toolkits in our codebase, including GPU memory usage (Figure 4), loss landscape (Figure 5(c)), Power Law (PL) exponent alpha metrics (Figure 6), robustness and calibration (Table A8), *etc*. For instance, despite the key role *static* mixing plays in today's deep learning systems, we surprisingly find that its generalizability over diverse datasets and backbones is significantly inferior to that of *dynamic* algorithms. By ranking the performance and efficiency trade-offs, we reveal that recent *dynamic* methods have already outperformed the *static* ones. This may suggest a promising breakthrough for mixup augmentation, provided that the *dynamic* computational overhead can be further reduced. Overall, we believe these insights can facilitate better evaluation and comparisons of mixup methods, enabling a systematic understanding and thus paving the way for further advancements.

Since such a first-of-its benchmark can be rather time- and resource-consuming and most current advances have focused on and stemmed from visual classification tasks, we centralize our benchmarking scope on classification while extending it to broader mixup applications with transfer learning.

Meanwhile, we have already supported these downstream tasks and datasets in our open-source codebase, allowing practitioners to customize their mixup algorithms, models, and training setups in these relatively under-explored scenarios. Our key contributions can be summarized as follows:

- We introduce OpenMixup, the *first* comprehensive benchmarking study for mixup augmentation, where 18 representative baselines are trained from scratch and rigorously evaluated on 11 visual classification datasets, ranging from non-iconic scenes to gray-scale, fine-grained, and long tail scenarios. By providing a standard testbed and a rich set of evaluation protocols, OpenMixup enables fair comparisons, thorough assessment, and analysis of different mixup strategies.

- To support reproducible mixup research and user-friendly method deployment, we provide an open-source codebase for visual representation learning. The codebase incorporates standardized modules for data pre-processing, mixup augmentation, backbone selection, optimization policies, and distributed training functionalities. Beyond the benchmark itself, our OpenMixup codebase is readily extensible and has supported semi- and self-supervised learning and visual attribute regression tasks, which further enhances its utility and potential benefits to the community.

- Observations and insights are obtained through extensive analysis. We investigate the generalization ability of all evaluated mixup baselines across diverse datasets and backbones, compare their GPU memory footprint and computational cost, visualize the loss landscape and PL exponent alpha metrics to understand optimization behavior, and evaluate robustness against input corruptions and calibration performance. Furthermore, we establish comprehensive rankings in terms of their performance and applicability (efficiency and versatility), offering clear method guidelines for specific requirements. These findings not only present a firm grasp of the current mixup augmentation landscape but shed light on promising avenues for future advancements.

## 2 BACKGROUND AND RELATED WORK

### 2.1 PROBLEM DEFINITION

**Mixup Training.** We first consider the general image classification tasks with $k$ different classes: given a finite set of $n$ image samples $X = [x_i]_{i=1}^n \in \mathbb{R}^{n \times W \times H \times C}$ and their corresponding ground-truth class labels $Y = [y_i]_{i=1}^n \in \mathbb{R}^{n \times k}$, encoded by a one-hot vector $y_i \in \mathbb{R}^k$. We attempt to seek the mapping from input data $x_i$ to its class label $y_i$ modeled through a deep neural network $f_\theta : x \longmapsto y$ with parameters $\theta$ by optimizing a classification loss $\ell(.)$, say the cross entropy (CE) loss,

$$\ell_{CE}(f_\theta(x), y) = -y \log f_\theta(x). \tag{1}$$

Then we consider the mixup classification task: given a sample mixing function $h$, a label mixing function $g$, and a mixing ratio $\lambda$ sampled from $Beta(\alpha, \alpha)$ distribution, we can generate the mixed data $X_{mix}$ with $x_{mix} = h(x_i, x_j, \lambda)$ and the mixed label $Y_{mix}$ with $y_{mix} = g(y_i, y_j, \lambda)$, where $\alpha$ is a hyper-parameter. Similarly, we learn $f_\theta : x_{mix} \longmapsto y_{mix}$ by the mixup cross-entropy (MCE) loss,

$$\ell_{MCE} = \lambda \ell_{CE}(f_\theta(x_{mix}), y_i) + (1 - \lambda)\ell_{CE}(f_\theta(x_{mix}), y_j). \tag{2}$$

**Mixup Reformulation.** Comparing Eq. (1) and Eq. (2), the mixup training has the following features: (1) extra mixup policies, $g$ and $h$, are required to generate $X_{mix}$ and $Y_{mix}$. (2) the classification performance of $f_\theta$ depends on the generation policy of mixup. Naturally, we can split the mixup task into two complementary sub-tasks: (i) mixed sample generation and (ii) mixup classification (learning objective). Notice that the sub-task (i) is subordinate to (ii) because the final goal is to obtain a stronger classifier. Therefore, from this perspective, we regard the mixup generation as an auxiliary task for the classification task. Since $g$ is generally designed as a linear interpolation, i.e., $g(y_i, y_j, \lambda) = \lambda y_i + (1 - \lambda)y_j$, $h$ becomes the key function to determine the performance of the model. Generalizing previous offline methods, we define a parametric mixup policy $h_\phi$ as the sub-task with another set of parameters $\phi$. The final goal is to optimize $\ell_{MCE}$ given $\theta$ and $\phi$ as:

$$\min_{\theta, \phi} \ell_{MCE}\Big(f_\theta\big(h_\phi(x_i, x_j, \lambda)\big), g(y_i, y_j, \lambda)\Big). \tag{3}$$

### 2.2 SAMPLE MIXING

Within the realm of visual classification, prior research has primarily concentrated on refining the sample mixing strategies rather than the label mixing ones. In this context, most sample mixing methods are categorized into two groups: *static* policies and *dynamic* policies, as presented in Table 1.

Table 1: Overview of all supported vision Mixup augmentation methods in OpenMixup. Note that Mixup and CutMix in label mixing indicate mixing the labels of two samples by linear interpolation or computing cut squares. The *Perf.*, *App.*, and *Overall* denote the performance, applicability, and overall rankings of all methods, which are derived from average rankings across baselines (view B.5).

| Method | Category | Publication | Sample Mixing | Label Mixing | Extra Cost | ViT only | Perf. | App. | Overall |
|---|---|---|---|---|---|---|---|---|---|
| Mixup (Zhang et al., 2018) | Static | ICLR'2018 | Hand-crafted Interpolation | Mixup | ✗ | ✗ | 15 | 1 | 10 |
| CutMix (Yun et al., 2019) | Static | ICCV'2019 | Hand-crafted Cutting | CutMix | ✗ | ✗ | 13 | 1 | 8 |
| DeiT (CutMix+Mixup) (Touvron et al., 2021) | Static | ICML'2021 | CutMix+Mixup | CutMix+Mixup | ✗ | ✗ | 7 | 1 | 3 |
| SmoothMix (ha Lee et al., 2020) | Static | CVPRW'2020 | Hand-crafted Cutting | CutMix | ✗ | ✗ | 18 | 1 | 13 |
| GridMix (Baek et al., 2021) | Static | PR'2021 | Hand-crafted Cutting | CutMix | ✗ | ✗ | 17 | 1 | 12 |
| ResizeMix (Qin et al., 2023) | Static | CVMJ'2023 | Hand-crafted Cutting | CutMix | ✗ | ✗ | 10 | 1 | 5 |
| ManifoldMix (Verma et al., 2019) | Static | ICML'2019 | Latent-space Mixup | Mixup | ✗ | ✗ | 14 | 1 | 9 |
| FMix (Harris et al., 2020) | Static | arXiv'2020 | Fourier-guided Cutting | CutMix | ✗ | ✗ | 16 | 1 | 11 |
| AttentiveMix (Walawalkar et al., 2020) | Static | ICASSP'2020 | Pretraining-guided Cutting | CutMix | ✓ | ✗ | 9 | 3 | 6 |
| SaliencyMix (Uddin et al., 2020) | Static | ICLR'2021 | Saliency-guided Cutting | CutMix | ✗ | ✗ | 11 | 1 | 6 |
| PuzzleMix (Kim et al., 2020) | Dynamic | ICML'2020 | Optimal-transported Cutting | CutMix | ✓ | ✗ | 8 | 4 | 6 |
| AlignMix (Venkataramanan et al., 2022) | Dynamic | CVPR'2022 | Optimal-transported Interpolation | CutMix | ✓ | ✗ | 12 | 2 | 8 |
| AutoMix (Liu et al., 2022d) | Dynamic | ECCV'2022 | End-to-end-learned Cutting | CutMix | ✓ | ✗ | 3 | 6 | 4 |
| SAMix (Li et al., 2021) | Dynamic | arXiv'2021 | End-to-end-learned Cutting | CutMix | ✓ | ✗ | 1 | 5 | 1 |
| AdAutoMix (Qin et al., 2024) | Dynamic | ICLR'2024 | End-to-end-learned Cutting | CutMix | ✓ | ✗ | 2 | 7 | 4 |
| TransMix (Chen et al., 2022) | Dynamic | CVPR'2022 | CutMix+Mixup | Attention-guided | ✗ | ✓ | 5 | 8 | 7 |
| SMMix (Chen et al., 2023) | Dynamic | ICCV'2023 | CutMix+Mixup | Attention-guided | ✗ | ✓ | 4 | 8 | 6 |
| DecoupledMix (Liu et al., 2022c) | Static | NeurIPS'2023 | Any Sample Mixing Policies | DecoupledMix | ✗ | ✗ | 6 | 1 | 2 |

**Static Policies.** The sample mixing procedure in all *static* policies is conducted in a *hand-crafted* manner. Mixup (Zhang et al., 2018) first generates artificially mixed data through the convex combination of two selected input samples and their associated one-hot labels. ManifoldMix variants (Verma et al., 2019; Faramarzi et al., 2020) extend the same technique to latent space for smoother feature mixing. Subsequently, CutMix (Yun et al., 2019) involves the random replacement of a certain rectangular region inside the input sample while concurrently employing Drop Patch throughout the mixing process. Inspired by CutMix, several researchers in the community have explored the use of saliency information (Uddin et al., 2020) to pilot mixing patches, while others have developed more complex *hand-crafted* sample mixing strategies (Harris et al., 2020; Baek et al., 2021).

**Dynamic Policies.** In contrast to *static* mixing, *dynamic* strategies are proposed to incorporate sample mixing into an adaptive optimization-based framework. PuzzleMix variants (Kim et al., 2020; 2021) introduce combinatorial optimization-based mixing policies in accordance with saliency maximization. SuperMix variants (Dabouei et al., 2021; Walawalkar et al., 2020) utilize pre-trained teacher models to compute smooth and optimized samples. Distinctively, AutoMix variants (Liu et al., 2022d; Li et al., 2021) reformulate the overall framework of sample mixing into an *online-optimizable* fashion where the model learns to generate the mixed samples in an end-to-end manner.

## 2.3 LABEL MIXING

Mixup (Zhang et al., 2018) and CutMix (Yun et al., 2019) are two widely-recognized label mixing techniques, both of which are *static*. Recently, there has been a notable emphasis among researchers on advancing label mixing approaches, which attain more favorable performance upon certain sample mixing policies. Based on Transformers, TransMix variants (Chen et al., 2022; Liu et al., 2022a; Choi et al., 2022; Chen et al., 2023) are proposed to utilize class tokens and attention maps to adjust the mixing ratio. A decoupled mixup objective (Liu et al., 2022c) is introduced to force models to focus on those hard mixed samples, which can be plugged into different sample mixing policies. Holistically, most existing studies strive for advanced sample mixing designs rather than label mixing.

## 2.4 OTHER APPLICATIONS

Recently, mixup augmentation also has shown promise in more vision applications, such as semi-supervised learning (Berthelot et al., 2019; Liu et al., 2022c), self-supervised pre-training (Kalantidis et al., 2020; Shen et al., 2022), and visual attribute regression (Wu et al., 2022; Bochkovskiy et al., 2020). Although these fields are not as extensively studied as classification, our OpenMixup codebase has been designed to support them by including the necessary task settings and datasets. Its modular and extensible architecture allows researchers and practitioners in the community to effortlessly adapt and extend their models to accommodate the specific requirements of these tasks, enabling them to quickly set up experiments without building the entire pipeline from scratch. Moreover, our codebase will be well-positioned to accelerate the development of future benchmarks, ultimately contributing to the advancement of mixup augmentation across a diversity of visual representation learning tasks.

## 3 OPENMIXUP

This section introduces our OpenMixup codebase framework and benchmark from four key aspects: supported methods and tasks, evaluation metrics, and experimental pipeline. OpenMixup provides a

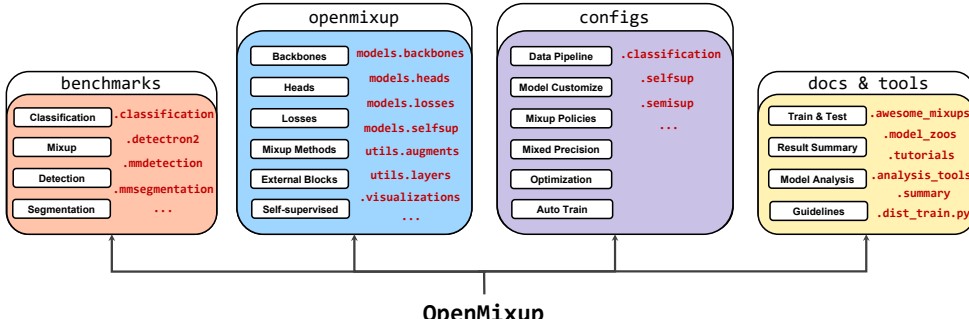

Figure 3: Overview of codebase framework of OpenMixup. (1) `benchmarks` provide benchmarking results and corresponding config files for mixup classification and transfer learning. (2) `openmixup` contains implementations of all supported methods. (3) `configs` is responsible for customizing setups of different mixup methods, networks, datasets, and training pipelines. (4) `docs & tools` contains paper lists of popular mixup methods, user documentation, and useful tools.

unified framework implemented in PyTorch (Paszke et al., 2019) for mixup model design, training, and evaluation. The framework references MMClassification (Contributors, 2020a) and follows the OpenMMLab coding style. We start with an overview of its composition. As shown in Figure 3, the whole training process here is fragmented into multiple components, including model architecture (`.openmixup.models`), data pre-processing (`.openmixup.datasets`), mixup policies (`.openmixup.models.utils.augments`), script tools (`.tools`) *etc.* For instance, vision models are summarized into modular building blocks (*e.g.*, backbone, neck, head *etc.*) in `.openmixup.models`. This modular architecture enables practitioners to easily craft models by incorporating different components through configuration files in `.configs`. As such, users can readily customize their specified vision models and training strategies. In addition, benchmarking configuration (`.benchmarks`) and results (`.tools.model_zoos`) are also provided in the codebase. Additional benchmarking configurations and details are discussed below.

## 3.1 BENCHMARKED METHODS

OpenMixup has implemented 17 representative mixup augmentation algorithms and 19 convolutional neural network and Transformer model architectures (gathered in `.openmixup.models`) across 12 diverse image datasets for supervised visual classification. We summarize these mixup methods in Table 1, along with their corresponding conference/journal, the types of employed sample, and label mixing policies, properties, and rankings. For sample mixing, Mixup (Zhang et al., 2018) and ManifoldMix (Verma et al., 2019) perform *hand-crafted* convex interpolation. CutMix (Yun et al., 2019), SmoothMix (ha Lee et al., 2020), GridMix (Baek et al., 2021) and ResizeMix (Qin et al., 2023) implement *hand-crafted* cutting policy. FMix (Harris et al., 2020) utilizes Fourier-guided cutting. AttentiveMix (Walawalkar et al., 2020) and SaliencyMix (Uddin et al., 2020) apply pretraining-guided and saliency-guided cutting, respectively. Some *dynamic* approaches like PuzzleMix (Kim et al., 2020) and AlignMix (Venkataramanan et al., 2022) utilize optimal transport-based cutting and interpolation. AutoMix (Liu et al., 2022d) and SAMix (Li et al., 2021) perform end-to-end online-optimizable cutting-based approaches. As for the label mixing, most methods apply Mixup (Zhang et al., 2018) or CutMix (Yun et al., 2019), while the latest mixup methods for visual transformers (TransMix (Chen et al., 2022), TokenMix (Liu et al., 2022a), and SMMix (Chen et al., 2023)), as well as DecoupledMix (Liu et al., 2022c) exploit attention maps and a decoupled framework respectfully instead, which incorporate CutMix variants as its sample mixing strategy. Such a wide scope of supported methods enables a comprehensive benchmarking analysis on visual classification.

## 3.2 BENCHMARKING TASKS

We provide detailed descriptions of the 12 open-source datasets as shown in Table 2. These datasets can be classified into four categories below: **(1) Small-scale classification**: We conduct benchmarking studies on small-scale datasets to provide an accessible benchmarking reference. CIFAR-10/100 (Krizhevsky et al., 2009) consists of 60,000 color images in 32×32 resolutions. Tiny-ImageNet (Tiny) (Chrabaszcz et al., 2017) and STL-10 (Coates et al., 2011) are two re-scale versions of ImageNet-1K in the size of 64×64 and 96×96. FashionMNIST (Xiao et al., 2017) is the advanced version of MNIST, which contains gray-scale images of clothing. **(2) Large-scale classification**: The large-scale dataset is employed to evaluate mixup algorithms against the most

Table 2: The detailed information of supported visual classification datasets in OpenMixup.

| Datasets | Category | Source | Classes | Resolution | Train images | Test images |
|---|---|---|---|---|---|---|
| CIFAR-10 (Krizhevsky et al., 2009) | Iconic | link | 10 | 32×32 | 50,000 | 10,000 |
| CIFAR-100 (Krizhevsky et al., 2009) | Iconic | link | 100 | 32×32 | 50,000 | 10,000 |
| FashionMNIST (Xiao et al., 2017) | Gray-scale | link | 10 | 28×28 | 50,000 | 10,000 |
| STL-10 (Coates et al., 2011) | Iconic | link | 10 | 96×96 | 50,00 | 8,000 |
| Tiny-ImageNet (Chrabaszcz et al., 2017) | Iconic | link | 200 | 64×64 | 10,000 | 10,000 |
| ImageNet-1K (Russakovsky et al., 2015) | Iconic | link | 1000 | 469×387 | 1,281,167 | 50,000 |
| CUB-200-2011 (Wah et al., 2011) | Fine-grained | link | 200 | 224×224 | 5,994 | 5,794 |
| FGVC-Aircraft (Maji et al., 2013) | Fine-grained | link | 100 | 224×224 | 6,667 | 3,333 |
| iNaturalist2017 Horn et al. (2018) | Fine-grained & longtail | link | 5089 | 224×224 | 579,184 | 95,986 |
| iNaturalist2018 Horn et al. (2018) | Fine-grained & longtail | link | 8142 | 224×224 | 437,512 | 24,426 |
| Places205 (Zhou et al., 2014) | Scenic | link | 205 | 224×224 | 2,448,873 | 41,000 |

standardized procedure, which can also support the prevailing ViT architecture. ImageNet-1K (IN-1K) (Russakovsky et al., 2015) is a well-known challenging dataset for image classification with 1000 classes. **(3) Fine-grained classification**: To investigate the effectiveness of mixup methods in complex inter-class relationships and long-tail scenarios, we conduct a comprehensive evaluation of fine-grained classification datasets, which can also be classified into small-scale and large-scale scenarios. (i) *Small-scale scenarios*: The datasets for small-scale fine-grained evaluation scenario are CUB-200-2011 (CUB) (Wah et al., 2011) and FGVC-Aircraft (Aircraft) (Maji et al., 2013), which contains a total of 200 wild bird species and 100 classes of airplanes. (ii) *Large-scale scenarios*: The datasets for large-scale fine-grained evaluation scenarios are iNaturalist2017 (iNat2017) (Horn et al., 2018) and iNaturalist2018 (iNat2018) (Horn et al., 2018), which contain 5,089 and 8,142 natural categories. Both the iNat2017 and iNat2018 own 7 major categories and are also long-tail datasets with scenic images (*i.e.*, the fore-ground target is within large backgrounds). **(4) Scenic classification**: Scenic classification evaluations are also conducted to investigate the performance of different mixup augmentation methods in complex non-iconic scenarios on Places205 (Zhou et al., 2014).

### 3.3 EVALUATION METRICS AND TOOLS

We comprehensively evaluate the beneficial properties of mixup augmentation algorithms on the aforementioned vision tasks through the use of various metrics and visualization analysis tools in a rigorous manner. Overall, the evaluation methodologies can be classified into two distinct divisions, namely performance metric and empirical analysis. For the performance metrics, classification accuracy and robustness against corruption are two performance indicators examined. As for empirical analysis, experiments on calibrations, CAM visualization, loss landscape, the plotting of training loss, and validation accuracy curves are conducted. The utilization of these approaches is contingent upon their distinct properties, enabling user-friendly deployment for designated purposes and demands.

**Performance Metric.** **(1) Accuracy and training costs**: We adopt top-1 accuracy, total training hours, and GPU memory to evaluate all mixup methods' classification performance and training costs. **(2) Robustness**: We evaluate the robustness against corruptions of the methods on CIFAR-100-C and ImageNet-C (Russakovsky et al., 2015), which is designed for evaluating the corruption robustness and provides 19 different corruptions, *e.g.*, noise and blur *etc.* **(3) Transferability to downstream tasks**: We evaluate the transferability of existing methods to object detection based on Faster R-CNN (Ren et al., 2015) and Mask R-CNN (He et al., 2017) on COCO *train2017* (Lin et al., 2014), initializing with trained models on ImageNet. We also transfer these methods to semantic segmentation on ADE20K (Zhou et al., 2018). Please refer to Appendix B.4 for details.

**Empirical Analysis.** **(1) Calibrations**: To verify the calibration of existing methods, we evaluate them by the expected calibration error (ECE) on CIFAR-100 (Krizhevsky et al., 2009), *i.e.*, the absolute discrepancy between accuracy and confidence. **(2) CAM visualization**: We utilize mixed sample visualization, a series of CAM variants (Chattopadhay et al., 2018; Muhammad & Yeasin, 2020) (*e.g.*, Grad-CAM (Selvaraju et al., 2019)) to directly analyze the classification accuracy and especially the localization capabilities of mixup augmentation algorithms through top-1 top-2 accuracy predicted targets. **(3) Loss landscape**: We apply loss landscape evaluation (Li et al., 2018) to further analyze the degree of loss smoothness of different mixup augmentation methods. **(4) Training loss and accuracy curve**: We plot the training losses and validation accuracy curves of various mixup methods to analyze the training stability, the ability to prevent over-fitting, and convergence speed. **(5) Quality metric of learned weights**: Employing WeightWatch (Martin et al., 2021), we plot the Power Law (PL) exponent alpha metric of learned parameters with mixup algorithms to study their properties on different scenarios, *e.g.,* acting as the regularizer to prevent overfitting or expanding more data as the augmentation technique to learn better representations.

Table 3: Top-1 accuracy (%) on CIFAR-10/100 and Tiny-ImageNet (Tiny) based on ResNet (R), Wide-ResNet (WRN), and ResNeXt (RX) backbones.

| Datasets | CIFAR-10 | CIFAR-100 | Tiny |
|---|---|---|---|
| Backbones | R-18 | WRN-28-8 | RX-50 |
| Epochs | 800 ep | 800 ep | 400 ep |
| Vanilla | 95.50 | 81.63 | 65.04 |
| Mixup | 96.62 | 82.82 | 66.36 |
| CutMix | 96.68 | 84.45 | 66.47 |
| ManifoldMix | 96.71 | 83.24 | 67.30 |
| SmoothMix | 96.17 | 82.09 | 68.61 |
| AttentiveMix | 96.63 | 84.34 | 67.42 |
| SaliencyMix | 96.20 | 84.35 | 66.55 |
| FMix | 96.18 | 84.21 | 65.08 |
| GridMix | 96.56 | 84.24 | 69.12 |
| ResizeMix | 96.76 | 84.87 | 65.87 |
| PuzzleMix | 97.10 | 85.02 | 67.83 |
| Co-Mixup | 97.15 | 85.05 | 68.02 |
| AlignMix | 97.05 | 84.87 | 68.74 |
| AutoMix | 97.34 | 85.18 | 70.72 |
| SAMix | 97.50 | **85.50** | 72.18 |
| AdAutoMix | **97.55** | 85.32 | **72.89** |
| Decoupled | 96.95 | 84.88 | 67.46 |

Table 4: Top-1 accuracy (%) on ImageNet-1K using PyTorch-style, RSB A2/A3, and DeiT settings based on CNN and Transformer architectures, including ResNet (R), MobileNet.V2 (Mob.V2), DeiT-S, and Swin-T.

| Backbones | R-50 | R-50 | Mob.V2 1x | DeiT-S | Swin-T |
|---|---|---|---|---|---|
| Epochs | 100 ep | 100 ep | 300 ep | 300 ep | 300 ep |
| Settings | PyTorch | RSB A3 | RSB A2 | DeiT | DeiT |
| Vanilla | 76.83 | 77.27 | 71.05 | 75.66 | 80.21 |
| Mixup | 77.12 | 77.66 | 72.78 | 77.72 | 81.01 |
| CutMix | 77.17 | 77.62 | 72.23 | 80.13 | 81.23 |
| DeiT / RSB | 77.35 | 78.08 | 72.87 | 79.80 | 81.20 |
| ManifoldMix | 77.01 | 77.78 | 72.34 | 78.03 | 81.15 |
| AttentiveMix | 77.28 | 77.46 | 70.30 | 80.32 | 81.29 |
| SaliencyMix | 77.14 | 77.93 | 72.07 | 79.88 | 81.37 |
| FMix | 77.19 | 77.76 | 72.79 | 80.45 | 81.47 |
| ResizeMix | 77.42 | 77.85 | 72.50 | 78.61 | 81.36 |
| PuzzleMix | 77.54 | 78.02 | 72.85 | 77.37 | 79.60 |
| AutoMix | 77.91 | 78.44 | 73.19 | 80.78 | 81.80 |
| SAMix | **78.06** | **78.64** | **73.42** | 80.94 | **81.87** |
| AdAutoMix | 78.04 | 78.54 | - | 80.81 | 81.75 |
| TransMix | - | - | - | 80.68 | 81.80 |
| SMMix | - | - | - | **81.10** | 81.80 |

## 3.4 Experimental Pipeline of OpenMixup Codebase

OpenMixup provides a unified training pipeline that offers a consistent workflow across various computer vision tasks, as illustrated in Figure A1. Taking image classification as an example, we can outline the overall training process as follows. (i) Data preparation: Users first select the appropriate dataset and pre-processing techniques from our supported data pipeline. (ii) Model architecture: The `openmixup.models` module serves as a component library for building desired model architectures. (iii) Configuration: Users can easily customize their experimental settings using Python configuration files under `.configs.classification`. These files allow for the specification of datasets, mixup strategies, neural networks, and schedulers. (iv) Execution: The `.tools` directory not only provides hardware support for distributed training but offers utility functionalities, such as feature visualization, model analysis, and result summarization, which can further facilitate empirical analysis. We also provide comprehensive online user documents, including detailed guidelines for installation and getting started instructions, all the benchmarking results, and awesome lists of related works in mixup augmentation, *etc.*, which ensures that both researchers and practitioners in the community can effectively leverage our OpenMixup for their specific needs.

## 4 Experiment and Analysis

### 4.1 Implementation Details

We conduct essential benchmarking experiments of image classification on various scenarios with diverse evaluation metrics. For a fair comparison, grid search is performed for the shared hyper-parameter $\alpha \in \{0.1, 0.2, 0.5, 1, 2, 4\}$ of supported mixup variants while the rest of the hyper-parameters follow the original papers. Vanilla denotes the classification baseline without any mixup augmentations. All experiments are conducted on Ubuntu workstations with Tesla V100 or NVIDIA A100 GPUs and report the *mean* results of three trials. Appendix B provides full visual classification results, Appendix B.4 presents our transfer learning results for object detection and semantic segmentation, and Appendix C conduct verification of the reproduction guarantee in OpenMixup.

**Small-scale Benchmarks.** We first provide standard mixup image classification benchmarks on five small datasets with two settings. (a) The classical settings with the CIFAR version of ResNet variants (He et al., 2016; Xie et al., 2017), *i.e.*, replacing the $7 \times 7$ convolution and MaxPooling by a $3 \times 3$ convolution. We use $32 \times 32$, $64 \times 64$, and $28 \times 28$ input resolutions for CIFAR-10/100, Tiny-ImageNet, and FashionMNIST, while using the normal ResNet for STL-10. We train vision models for multiple epochs from the stretch with SGD optimizer and a batch size of 100, as shown in Table 3 and Appendix B.2. (b) The modern training settings following DeiT (Touvron et al., 2021) on CIFAR-100, using $224 \times 224$ and $32 \times 32$ resolutions for Transformers (DeiT-S (Touvron et al., 2021) and Swin-T (Liu et al., 2021)) and ConvNeXt-T (Liu et al., 2022b) as shown in Table A7.

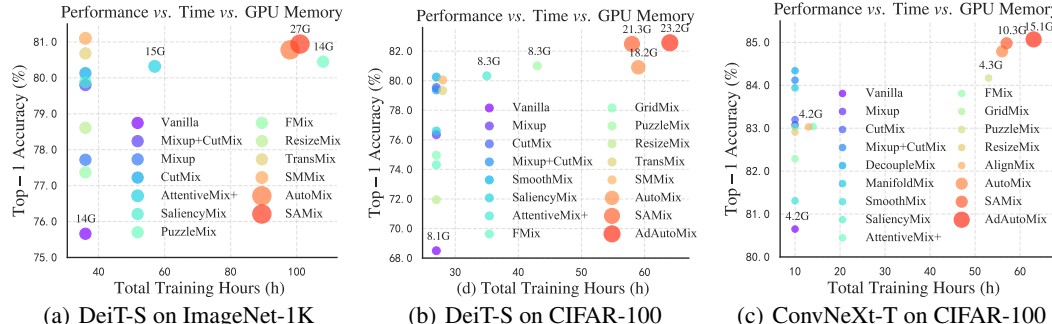

(a) DeiT-S on ImageNet-1K    (b) DeiT-S on CIFAR-100    (c) ConvNeXt-T on CIFAR-100

Figure 4: Trade-off evaluation with respect to accuracy performance, total training time (hours), and GPU memory (G). The results in (a) are based on DeiT-S architecture on ImageNet-1K. The results in (b) and (c) are based on DeiT-S and ConvNeXt-T backbones on CIFAR-100, respectively.

Table 5: Rankings of various mixup augmentations as take-home messages for practical usage.

|  | Mixup | CutMix | DeiT | SmoothMix | GridMix | ResizeMix | ManifoldMix | FMix | AttentiveMix | SaliencyMix | PuzzleMix | AlignMix | AutoMix | SAMix | TransMix | SMMix |
|---|---|---|---|---|---|---|---|---|---|---|---|---|---|---|---|---|
| Performance | 13 | 11 | 5 | 16 | 15 | 8 | 12 | 14 | 7 | 9 | 6 | 10 | 2 | 1 | 4 | 3 |
| Applicability | 1 | 1 | 1 | 1 | 1 | 1 | 1 | 1 | 3 | 1 | 4 | 2 | 7 | 6 | 5 | 5 |
| Overall | 8 | 6 | 1 | 11 | 10 | 4 | 7 | 9 | 5 | 5 | 5 | 6 | 4 | 2 | 4 | 3 |

**Standard ImageNet-1K Benchmarks.** For visual augmentation and network architecture communities, ImageNet-1K is a well-known standard dataset. We support three popular training recipes: (a) PyTorch-style (He et al., 2016) setting for classical CNNs; (b) timm RSB A2/A3 (Wightman et al., 2021) settings; (c) DeiT (Touvron et al., 2021) setting for ViT-based models. Evaluation is performed on 224×224 resolutions with CenterCrop. Popular network architectures are considered: ResNet (He et al., 2016), Wide-ResNet (Zagoruyko & Komodakis, 2016), ResNeXt (Xie et al., 2017), MobileNet.V2 (Sandler et al., 2018), EfficientNet (Tan & Le, 2019), DeiT (Touvron et al., 2021), Swin (Liu et al., 2021), ConvNeXt (Liu et al., 2022b), and MogaNet (Li et al., 2024). Refer to Appendix A for implementation details. In Table 4 and Table A2, we report the *mean* performance of three trials where the *median* of top-1 test accuracy in the last 10 epochs is recorded for each trial.

**Benchmarks on Fine-grained and Scenic Scenarios.** We further provide benchmarking results on three downstream classification scenarios in 224×224 resolutions with ResNet backbone architectures: (a) Transfer learning on CUB-200 and FGVC-Aircraft. (b) Fine-grained classification on iNat2017 and iNat2018. (c) Scenic classification on Places205, as illustrated in Appendix B.3 and Table A10.

### 4.2 Observations and Insights

Empirical analysis is conducted to gain insightful observations and a systematic understanding of the properties of different mixup augmentation techniques. Our key findings are summarized as follows:

**(A) Which mixup method should I choose?** Integrating benchmarking results from various perspectives, we provide practical mixup rankings (detailed in Appendix B.5) as a take-home message for real-world applications, which regards performance, applicability, and overall capacity. As shown in Table 1, as for the performance, the *online-optimizable* SAMix and AutoMix stand out as the top two choices. SMMix and TransMix follow closely behind. However, regarding applicability that involves both the concerns of efficiency and versatility, *hand-crafted* methods significantly outperform the learning-based ones. Overall, the DeiT (Mixup+CutMix), SAMix, and SMMix are selected as the three most preferable mixup methods, each with its own emphasis. Table 5 shows ranking results.

**(B) Generalization over datasets.** The intuitive performance radar chart presented in Figure 1, combined with the trade-off results in Figure 4, reveals that *dynamic* mixup methods consistently yield better performance compared to *static* ones, showcasing their impressive generalizability. However, *dynamic* approaches necessitate meticulous tuning, which incurs considerable training costs. In contrast, *static* mixup exhibits significant performance fluctuation across different datasets, indicating poor generalizability with application scenarios. For instance, Mixup and CutMix as the *static* representatives perform even worse than the baseline on Place205 and FGVC-Aircraft, respectively. Moreover, we analyze how mixup methods improve on different datasets in Figure 6 and Figure A4. On small-scale datasets, mixup methods (*dynamic* ones) tend to prevent the over-parameterized

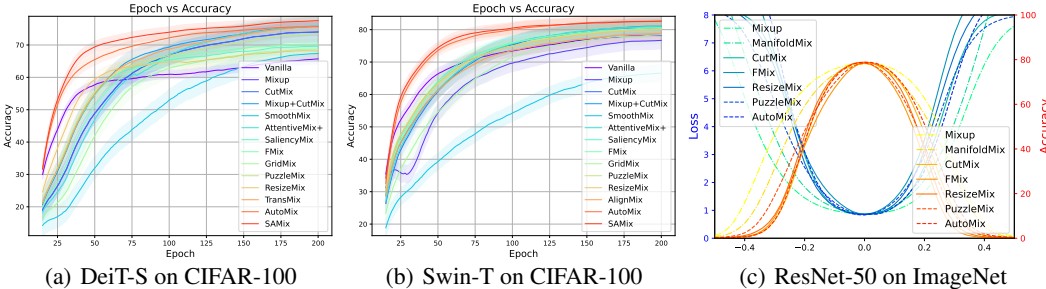

(a) DeiT-S on CIFAR-100     (b) Swin-T on CIFAR-100     (c) ResNet-50 on ImageNet

Figure 5: (a)(b) Training epoch *vs.* top-1 accuracy (%) plots of different mixup methods on CIFAR-100 to analyze training stability and convergence speed. (c) 1-D loss landscapes for mixup methods with ResNet-50 (300 epochs) on ImageNet-1K. The results show that *dynamic* approaches achieve deeper and wider loss landscapes than *static* ones, which may indicate better optimization behavior.

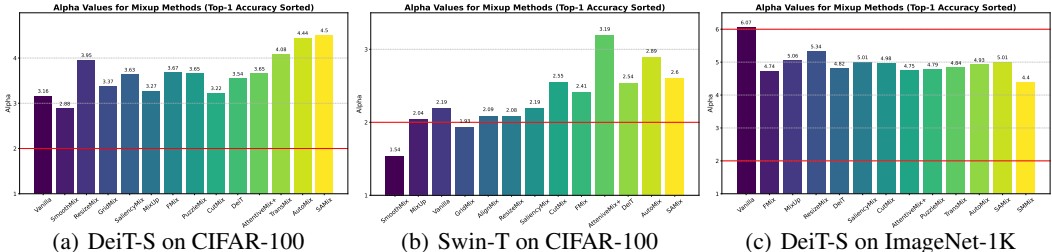

(a) DeiT-S on CIFAR-100     (b) Swin-T on CIFAR-100     (c) DeiT-S on ImageNet-1K

Figure 6: Visualization of PL exponent alpha metrics (Martin et al., 2021) of learned models by different mixup based on DeiT-S or Swin-T on (a)(b) CIFAR-100 and (c) ImageNet-1K. In each figure, the bars are sorted with the top-1 accuracy from left to right. Holistically, the alpha metric measures the fitting degree of the learned model to a certain task. A smaller alpha indicates better task fitting. Empirically, values less than 2 or larger than 6 run the risk of overfitting and underfitting. Therefore, this could serve as a favorable toolkit to evaluate the impact of different mixups on models.

backbones (Vanilla or with some *static* ones) from overfitting. On the contrary, mixup techniques are served as data augmentations to encourage the model to fit hard tasks on large-scale datasets.

**(C) Generalization over backbones.** As shown in Figure 4 and Figure 5(c), we provide extensive evaluations on ImageNet-1K based on different types of backbones and mixup methods. As a result, *dynamic* mixup achieves better performance in general and shows more favorable generaliz-ability against backbone selection compared to *static* methods. Noticeably, the *online-optimizable* SAMix and AutoMix exhibit impressive generalization ability over different vision backbones, which potentially reveals the superiority of their online training framework compared to the others.

**(D) Applicability.** Figure A2 shows that ViT-specific methods (*e.g.*, TransMix (Chen et al., 2022) and TokenMix (Liu et al., 2022a)) yield exceptional performance with DeiT-S and PVT-S yet exhibit intense sensitivity to different model scales (*e.g.*, with PVT-T). Moreover, they are limited to ViTs, which largely restricts their applicability. Surprisingly, *static* Mixup (Zhang et al., 2018) exhibits favorable applicability with new efficient networks like MogaNet (Li et al., 2024). CutMix (Yun et al., 2019) fits well with popular backbones, such as modern CNNs (*e.g.*, ConvNeXt and ResNeXt) and DeiT, which increases its applicability. As shown in Figure 4, although AutoMix and SAMix are available in both CNNs and ViTs with consistent superiority, they have limitations in GPU memory and training time, which may limit their applicability in certain cases. This also provides a promising avenue for reducing the cost of well-performed online learnable mixup augmentation algorithms.

**(E) Robustness & Calibration.** We evaluate the robustness with accuracy on the corrupted version of CIFAR-100 and FGSM attack (Goodfellow et al., 2015) and the prediction calibration. Table A8 shows that all the benchmarked methods can improve model robustness against corruptions. However, only four recent *dynamic* approaches exhibit improved robustness compared to the baseline with FGSM attacks. We thus hypothesize that the *online-optimizable* mixup methods are robust against human interference, while the *hand-crafted* ones adapt to natural disruptions like corruption but are susceptible to attacks. Overall, AutoMix and SAMix achieve the optimal robustness and calibration results. For scenarios where these properties are required, practitioners can prioritize these methods.

**(F) Convergence & Training Stability.** As shown in Figure 5, wider bump curves indicate smoother loss landscapes (*e.g.*, Mixup), while higher warm color bump tips are associated with better conver-

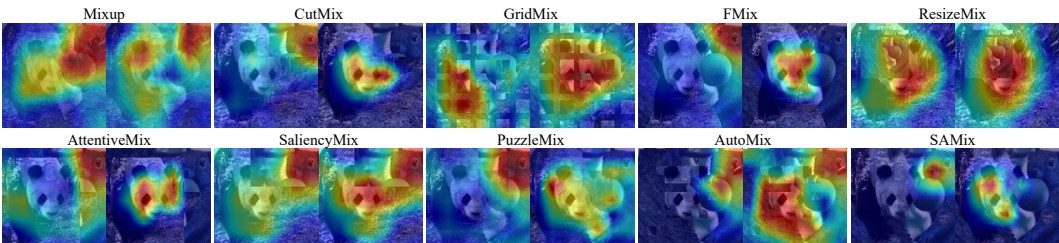

Figure 7: Visualization of class activation mapping (CAM) (Selvaraju et al., 2019) for top-1 and top-2 predicted classes of supported mixup methods with ResNet-50 on ImageNet-1K. Comparing the first and second rows, we observe that saliency-guided or dynamic mixup approaches (*e.g.*, PuzzleMix and SAMix) localize the target regions better than the static methods (*e.g.*, Mixup and ResizeMix).

gence and performance (*e.g.*, AutoMix). Evidently, *dynamic* mixup algorithms own better training stability and convergence than *static* mixup in general while obtaining sharp loss landscapes. They are likely to improve performances through exploring hard mixup samples. Nevertheless, the *static* mixup variants with convex interpolation, especially vanilla Mixup, exhibit smoother loss landscape and stable training than some *static* cutting-based methods. Based on the observations, we assume this arises from its interpolation that prioritizes training stability but may lead to sub-optimal results.

**(G) Downstream Transferability & CAM Visualization.** To further evaluate the downstream performance and transferability of different mixup methods, we conduct transfer learning experiments on object detection (Ren et al., 2015), semantic segmentation (Kirillov et al., 2019), and weakly supervised object localization (Choe et al., 2020) with details in Appendix B.4. Notably, Table A11, Table A12, and Table A13 suggest that *dynamic* sampling mixing methods like AutoMix indeed exhibit competitive results, while recently proposed ViT-specific label mixing methods like TransMix perform even better, showcasing their superior transferability. The results also show the potential for improved online training mixup design. Moreover, it is commonly conjectured that vision models with better CAM localization could potentially be better transferred to fine-grained downstream prediction tasks. As such, to gain intuitive insights, we also provide tools for class activation mapping (CAM) visualization with predicted classes in our codebase. As shown in Figure 7 and Table A13, *dynamic* mixup like SAMix and AutoMix shows exceptional CAM localization, combined with their aforementioned accuracy, generalization ability, and robustness, may indicate their practical superiority compared to the *static* ones in object detection and even borader downstream tasks.

## 5 CONCLUSION AND DISCUSSION

**Contributions.** This paper presents OpenMixup, the *first* comprehensive mixup augmentation benchmark and open-source codebase for visual representation learning, where 18 mixup algorithms are trained and thoroughly evaluated on 11 diverse vision datasets. The released codebase not only bolsters the entire benchmark but can facilitate broader under-explored mixup applications and downstream tasks. Furthermore, observations and insights are obtained through different aspects of empirical analysis that are previously under-explored, such as GPU memory usage, loss landscapes, PL exponent alpha metrics, and more, contributing to a deeper and more systematic comprehension of mixup augmentation. We anticipate that our OpenMixup benchmark and codebase can further contribute to fair and reproducible mixup research and we also encourage researchers and practitioners in the community to extend their valuable feedback to us and contribute to OpenMixup for building a more constructive mixup-based visual representation learning codebase together through GitHub.

**Limitations and Future Works.** The benchmarking scope of this work mainly focuses on visual classification, albeit we have supported a broader range of tasks in the proposed codebase and have conducted transfer learning experiments to object detection and semantic segmentation tasks to draw preliminary conclusions. We are aware of this and have prepared it upfront for future work. For example, our codebase can be easily extended to other computer vision tasks and datasets for further mixup benchmarking experiments and evaluations if necessary. Moreover, our observations and insights can also be of great value to the community for a more comprehensive understanding of mixup augmentation techniques. We believe this work as the *first* mixup benchmarking study is enough to serve as a kick-start, and we plan to extend our work in these directions in the future.

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

## SUPPLEMENT MATERIAL

In supplement material, we provide implementation details and full benchmark results of image classification, downstream tasks, and empirical analysis with mixup augmentations implemented in `OpenMixup` on various datasets.

## A  IMPLEMENTATION DETAILS

### A.1  SETUP OPENMIXUP

As provided in the supplementary material or the `online document`, we simply introduce the installation and data preparation for OpenMixup, detailed in "docs/en/latest/install.md". Assuming the PyTorch environment has already been installed, users can easily reproduce the environment with the source code by executing the following commands:

```
conda activate openmixup
pip install openmim
mim install mmcv-full
\# put the source code here
cd openmixup
python setup.py develop  \# or "pip install -e ."
```

Executing the instructions above, OpenMixup will be installed as the development mode, *i.e.*, any modifications to the local source code take effect, and can be used as a python package. Then, users can download the datasets and the released meta files and symlink them to the dataset root (`$OpenMixup/data`). The codebase is under `Apache 2.0` license.

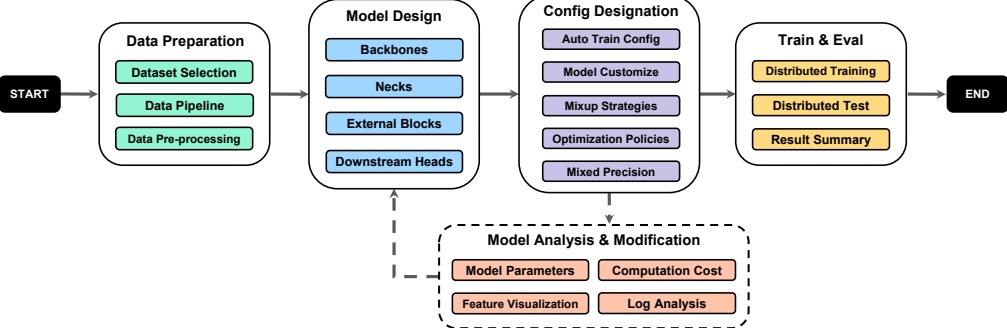

Figure A1: Overview of the experimental pipeline in OpenMixup codebase.

### A.2  TRAINING SETTINGS OF IMAGE CLASSIFICATION

**Large-scale Datasets.**  Table A1 illustrates three popular training settings on large-scaling datasets like ImageNet-1K in detail: (1) PyTorch-style (Paszke et al., 2019). (2) DeiT (Touvron et al., 2021). (3) RSB A2/A3 (Wightman et al., 2021). Notice that the step learning rate decay strategy is replaced by Cosine Scheduler (Loshchilov & Hutter, 2016), and `ColorJitter` as well as `PCA lighting` are removed in PyTorch-style setting for better performances. DeiT and RSB settings adopt advanced augmentation and regularization techniques for Transformers, while RSB A3 is a simplified setting for fast training on ImageNet-1K. For a fare comparison, we search the optimal hyper-parameter $\alpha$ in $Beta(\alpha, \alpha)$ from $\{0.1, 0.2, 0.5, 1, 2, 4\}$ for compared methods while the rest of the hyper-parameters follow the original papers.

**Small-scale Datasets.**  We also provide two experimental settings on small-scale datasets: (a) Following the common setups (He et al., 2016; Yun et al., 2019) on small-scale datasets like CIFAR-10/100, we train 200/400/800/1200 epochs from stretch based on CIFAR version of ResNet variants (He et al., 2016), *i.e.*, replacing the $7 \times 7$ convolution and MaxPooling by a $3 \times 3$ convolution. As for the data augmentation, we apply `RandomFlip` and `RandomCrop` with 4 pixels padding for 32×32 resolutions. The testing image size is 32×32 (no `CenterCrop`). The basic training settings include: SGD optimizer with SGD weight decay of 0.0001, a momentum of 0.9, a batch size of 100, and a basic learning rate is 0.1 adjusted by Cosine Scheduler (Loshchilov & Hutter, 2016). (b) We also provide modern training settings following DeiT (Touvron et al., 2021), while using $224 \times 224$

Table A1: Ingredients and hyper-parameters used for ImageNet-1K training settings.

| Procedure | PyTorch | DeiT | RSB A2 | RSB A3 |
|---|---|---|---|---|
| Train Res | 224 | 224 | 224 | 160 |
| Test Res | 224 | 224 | 224 | 224 |
| Test crop ratio | 0.875 | 0.875 | 0.95 | 0.95 |
| Epochs | 100/300 | 300 | 300 | 100 |
| Batch size | 256 | 1024 | 2048 | 2048 |
| Optimizer | SGD | AdamW | LAMB | LAMB |
| LR | 0.1 | $1 \times 10^{-3}$ | $5 \times 10^{-3}$ | $8 \times 10^{-3}$ |
| LR decay | cosine | cosine | cosine | cosine |
| Weight decay | $10^{-4}$ | 0.05 | 0.02 | 0.02 |
| Warmup epochs | ✗ | 5 | 5 | 5 |
| Label smoothing $\epsilon$ | ✗ | 0.1 | ✗ | ✗ |
| Dropout | ✗ | ✗ | ✗ | ✗ |
| Stoch. Depth | ✗ | 0.1 | 0.05 | ✗ |
| Repeated Aug | ✗ | ✓ | ✓ | ✗ |
| Gradient Clip. | ✗ | 1.0 | ✗ | ✗ |
| H. flip | ✓ | ✓ | ✓ | ✓ |
| RRC | ✓ | ✓ | ✓ | ✓ |
| Rand Augment | ✗ | 9/0.5 | 7/0.5 | 6/0.5 |
| Auto Augment | ✗ | ✗ | ✗ | ✗ |
| Mixup alpha | ✗ | 0.8 | 0.1 | 0.1 |
| Cutmix alpha | ✗ | 1.0 | 1.0 | 1.0 |
| Erasing prob. | ✗ | 0.25 | ✗ | ✗ |
| ColorJitter | ✗ | ✗ | ✗ | ✗ |
| EMA | ✗ | ✓ | ✗ | ✗ |
| CE loss | ✓ | ✓ | ✗ | ✗ |
| BCE loss | ✗ | ✗ | ✓ | ✓ |
| Mixed precision | ✗ | ✗ | ✓ | ✓ |

Table A2: Top-1 accuracy (%) of image classification based on ResNet variants on ImageNet-1K using PyTorch-style 100-epoch and 300-epoch training procedures.

| Methods | Beta $\alpha$ | PyTorch 100 epochs | | | | | PyTorch 300 epochs | | | |
|---|---|---|---|---|---|---|---|---|---|---|
| | | R-18 | R-34 | R-50 | R-101 | RX-101 | R-18 | R-34 | R-50 | R-101 |
| Vanilla | - | 70.04 | 73.85 | 76.83 | 78.18 | 78.71 | 71.83 | 75.29 | 77.35 | 78.91 |
| MixUp | 0.2 | 69.98 | 73.97 | 77.12 | 78.97 | 79.98 | 71.72 | 75.73 | 78.44 | 80.60 |
| CutMix | 1 | 68.95 | 73.58 | 77.17 | 78.96 | 80.42 | 71.01 | 75.16 | 78.69 | 80.59 |
| ManifoldMix | 0.2 | 69.98 | 73.98 | 77.01 | 79.02 | 79.93 | 71.73 | 75.44 | 78.21 | 80.64 |
| SaliencyMix | 1 | 69.16 | 73.56 | 77.14 | 79.32 | 80.27 | 70.21 | 75.01 | 78.46 | 80.45 |
| FMix | 1 | 69.96 | 74.08 | 77.19 | 79.09 | 80.06 | 70.30 | 75.12 | 78.51 | 80.20 |
| ResizeMix | 1 | 69.50 | 73.88 | 77.42 | 79.27 | 80.55 | 71.32 | 75.64 | 78.91 | 80.52 |
| PuzzleMix | 1 | 70.12 | 74.26 | 77.54 | 79.43 | 80.53 | 71.64 | 75.84 | 78.86 | 80.67 |
| AutoMix | 2 | 70.50 | 74.52 | 77.91 | 79.87 | 80.89 | 72.05 | 76.10 | 79.25 | 80.98 |
| AdAutoMix | 1 | 70.86 | 74.82 | 78.04 | 79.91 | **81.09** | - | - | - | - |
| SAMix | 2 | **70.83** | **74.95** | **78.06** | **80.05** | 80.98 | **72.27** | **76.28** | **79.39** | **81.10** |

and $32 \times 32$ resolutions for Transformer and CNN architectures. We only changed the batch size to 100 for CIFAR-100 and borrowed other settings the same as DeiT on ImageNet-1K.

# B  MIXUP IMAGE CLASSIFICATION BENCHMARKS

## B.1  MIXUP BENCHMARKS ON IMAGENET-1K

**PyTorch-style training settings**  The benchmark results are illustrated in Table A2. Notice that we adopt $\alpha = 0.2$ for some cutting-based mixups (CutMix, SaliencyMix, FMix, ResizeMix) based on ResNet-18 since ResNet-18 might be under-fitted on ImageNet-1k.

**DeiT training setting**  Table A3 shows the benchmark results following DeiT training setting. Experiment details refer to Sec. A.2. Notice that the performances of transformer-based architectures are more difficult to reproduce than ResNet variants, and the mean of the best performance in 3 trials is reported as their original paper.

**RSB A2/A3 training settings**  The RSB A2/A3 benchmark results based on ResNet-50, EfficientNet-B0, and MobileNet.V2 are illustrated in Table A4. Training 300/100 epochs with the BCE loss on ImageNet-1k, RSB A3 is a fast training setting, while RSB A2 can exploit the full representation

Table A3: Top-1 accuracy (%) on ImageNet-1K based on popular Transformer-based architectures using DeiT-S training settings. Notice that † denotes reproducing results with the official implementation, while other results are implemented with OpenMixup. TransMix, TokenMix, and SMMix are specially designed for Transformers.

| Methods | $\alpha$ | DeiT-T | DeiT-S | DeiT-B | PVT-T | PVT-S | Swin-T | ConvNeXt-T | MogaNet-T |
|---|---|---|---|---|---|---|---|---|---|
| Vanilla | - | 73.91 | 75.66 | 77.09 | 74.67 | 77.76 | 80.21 | 79.22 | 79.25 |
| DeiT | 0.8, 1 | 74.50 | 79.80 | 81.83 | 75.10 | 78.95 | 81.20 | 82.10 | 79.02 |
| MixUp | 0.2 | 74.69 | 77.72 | 78.98 | 75.24 | 78.69 | 81.01 | 80.88 | 79.29 |
| CutMix | 0.2 | 74.23 | 80.13 | 81.61 | 75.53 | 79.64 | 81.23 | 81.57 | 78.37 |
| ManifoldMix | 0.2 | - | - | - | - | - | - | 80.57 | 79.07 |
| AttentiveMix+ | 2 | 74.07 | 80.32 | 82.42 | 74.98 | 79.84 | 81.29 | 81.14 | 77.53 |
| SaliencyMix | 0.2 | 74.17 | 79.88 | 80.72 | 75.71 | 79.69 | 81.37 | 81.33 | 78.74 |
| FMix | 0.2 | 74.41 | 77.37 | | 75.28 | 78.72 | 79.60 | 81.04 | 79.05 |
| ResizeMix | 1 | 74.79 | 78.61 | 80.89 | 76.05 | 79.55 | 81.36 | 81.64 | 78.77 |
| PuzzleMix | 1 | 73.85 | 80.45 | 81.63 | 75.48 | 79.70 | 81.47 | 81.48 | 78.12 |
| AutoMix | 2 | 75.52 | 80.78 | 82.18 | 76.38 | 80.64 | 81.80 | 82.28 | 79.43 |
| SAMix | 2 | **75.83** | **80.94** | 82.85 | **76.60** | 80.78 | **81.87** | **82.35** | **79.62** |
| TransMix | 0.8, 1 | 74.56 | 80.68 | 82.51 | 75.50 | 80.50 | 81.80 | - | - |
| TokenMix† | 0.8, 1 | 75.31 | 80.80 | **82.90** | 75.60 | - | 81.60 | - | - |
| SMMix | 0.8, 1 | 75.56 | 81.10 | 82.90 | 75.60 | **81.03** | 81.80 | - | - |

Table A4: Top-1 accuracy (%) on ImageNet-1K based on classical ConvNets using RSB A2/A3 training settings, including ResNet, EfficientNet, and MobileNet.V2.

| Backbones Settings | $Beta$ $\alpha$ | R-50 A3 | R-50 A2 | Eff-B0 A3 | Eff-B0 A2 | Mob.V2 1× A3 | Mob.V2 1× A2 |
|---|---|---|---|---|---|---|---|
| RSB | 0.1, 1 | 78.08 | 79.80 | 74.02 | 77.26 | 69.86 | 72.87 |
| MixUp | 0.2 | 77.66 | 79.39 | 73.87 | 77.19 | 70.17 | 72.78 |
| CutMix | 0.2 | 77.62 | 79.38 | 73.46 | 77.24 | 69.62 | 72.23 |
| ManifoldMix | 0.2 | 77.78 | 79.47 | 73.83 | 77.22 | 70.05 | 72.34 |
| AttentiveMix+ | 2 | 77.46 | 79.34 | 72.16 | 75.95 | 67.32 | 70.30 |
| SaliencyMix | 0.2 | 77.93 | 79.42 | 73.42 | 77.67 | 69.69 | 72.07 |
| FMix | 0.2 | 77.76 | 79.05 | 73.71 | 77.33 | 70.10 | 72.79 |
| ResizeMix | 1 | 77.85 | 79.94 | 73.67 | 77.27 | 69.94 | 72.50 |
| PuzzleMix | 1 | 78.02 | 79.78 | 74.10 | 77.35 | 70.04 | 72.85 |
| AutoMix | 2 | 78.44 | 80.28 | 74.61 | 77.58 | 71.16 | 73.19 |
| SAMix | 2 | **78.64** | **80.40** | **75.28** | **77.69** | **71.24** | **73.42** |

ability of ConvNets. Notice that the RSB settings employ Mixup with $\alpha = 0.1$ and CutMix with $\alpha = 1.0$. We report the mean of top-1 accuracy in the last 5/10 training epochs for 100/300 epochs.

### B.2 SMALL-SCALE CLASSIFICATION BENCHMARKS

To facilitate fast research on mixup augmentations, we benchmark mixup image classification on CIFAR-10/100 and Tiny-ImageNet with two settings.

**CIFAR-10** As elucidated in Sec. A.2, CIFAR-10 benchmarks based on CIFAR version ResNet variants follow CutMix settings, training 200/400/800/1200 epochs from stretch. As shown in Table A5, we report the median of top-1 accuracy in the last 10 training epochs.

**CIFAR-100** As for the classical setting (a), CIFAR-100 benchmarks train 200/400/800/1200 epochs from the stretch in Table A6, similar to CIFAR-10. Notice that we set weight decay to 0.0005 for cutting-based methods (CutMix, AttentiveMix+, SaliencyMix, FMix, ResizeMix) for better performances when using ResNeXt-50 (32x4d) as the backbone. As shown in Table A7 using the modern setting (b), we train three modern architectures for 200/600 epochs from the stretch. We resize the raw images to $224 \times 224$ resolutions for DeiT-S and Swin-T while modifying the stem network as the CIFAR version of ResNet for ConvNeXt-T with $32 \times 32$ resolutions. As shown in Table A8, we further provided more metrics to evaluate the robustness and reliability (Naseer et al., 2021; Song et al., 2023): evaluating top-1 accuracy on the corrupted version of CIFAR-100 (Hendrycks & Dietterich, 2019), applying FGSM attack (Goodfellow et al., 2015)), and visualizing the prediction calibration (Verma et al., 2019).

**Tiny-ImageNet** We largely follow the training setting of PuzzleMix (Kim et al., 2020) on Tiny-ImageNet, which adopts the basic augmentations of RandomFlip and RandomResizedCrop

Table A5: Top-1 accuracy (%) on CIFAR-10 training 200, 400, 800, 1200 epochs based on ResNet (R) and ResNeXt-32x4d (RX).

| Backbones | Beta | R-18 | R-18 | R-18 | R-18 | Beta | RX-50 | RX-50 | RX-50 | RX-50 |
| Epochs | α | 200 ep | 400 ep | 800 ep | 1200ep | α | 200 ep | 400 ep | 800 ep | 1200ep |
|---|---|---|---|---|---|---|---|---|---|---|
| Vanilla | - | 94.87 | 95.10 | 95.50 | 95.59 | - | 95.92 | 95.81 | 96.23 | 96.26 |
| MixUp | 1 | 95.70 | 96.55 | 96.62 | 96.84 | 1 | 96.88 | 97.19 | 97.30 | 97.33 |
| CutMix | 0.2 | 96.11 | 96.13 | 96.68 | 96.56 | 0.2 | 96.78 | 96.54 | 96.60 | 96.35 |
| ManifoldMix | 2 | 96.04 | 96.57 | 96.71 | 97.02 | 2 | 96.97 | 97.39 | 97.33 | 97.36 |
| SmoothMix | 0.5 | 95.29 | 95.88 | 96.17 | 96.17 | 0.2 | 95.87 | 96.37 | 96.49 | 96.77 |
| AttentiveMix+ | 2 | 96.21 | 96.45 | 96.63 | 96.49 | 2 | 96.84 | 96.91 | 96.87 | 96.62 |
| SaliencyMix | 0.2 | 96.05 | 96.42 | 96.20 | 96.18 | 0.2 | 96.65 | 96.89 | 96.70 | 96.60 |
| FMix | 0.2 | 96.17 | 96.53 | 96.18 | 96.01 | 0.2 | 96.72 | 96.76 | 96.76 | 96.10 |
| GridMix | 0.2 | 95.89 | 96.33 | 96.56 | 96.58 | 0.2 | 97.18 | 97.30 | 96.40 | 95.79 |
| ResizeMix | 1 | 96.16 | 96.91 | 96.76 | 97.04 | 1 | 97.02 | 97.38 | 97.21 | 97.36 |
| PuzzleMix | 1 | 96.42 | 96.87 | 97.10 | 97.13 | 1 | 97.05 | 97.24 | 97.37 | 97.34 |
| AutoMix | 2 | 96.59 | 97.08 | 97.34 | 97.30 | 2 | 97.19 | 97.42 | 97.65 | 97.51 |
| SAMix | 2 | **96.67** | **97.16** | **97.50** | **97.41** | 2 | **97.23** | **97.51** | **97.93** | **97.74** |

Table A6: Top-1 accuracy (%) on CIFAR-100 training 200, 400, 800, 1200 epochs based on ResNet (R), Wide-ResNet (WRN), ResNeXt-32x4d (RX). Notice that † denotes reproducing results with the official implementation, while other results are implemented with OpenMixup.

| Backbones | Beta | R-18 | R-18 | R-18 | R-18 | RX-50 | RX-50 | RX-50 | RX-50 | WRN-28-8 |
| Epochs | α | 200 ep | 400 ep | 800 ep | 1200ep | 200 ep | 400 ep | 800 ep | 1200ep | 400ep |
|---|---|---|---|---|---|---|---|---|---|---|
| Vanilla | - | 76.42 | 77.73 | 78.04 | 78.55 | 79.37 | 80.24 | 81.09 | 81.32 | 81.63 |
| MixUp | 1 | 78.52 | 79.34 | 79.12 | 79.24 | 81.18 | 82.54 | 82.10 | 81.77 | 82.82 |
| CutMix | 0.2 | 79.45 | 79.58 | 78.17 | 78.29 | 81.52 | 78.52 | 78.32 | 77.17 | 84.45 |
| ManifoldMix | 2 | 79.18 | 80.18 | 80.35 | 80.21 | 81.59 | 82.56 | 82.88 | 83.28 | 83.24 |
| SmoothMix | 0.2 | 77.90 | 78.77 | 78.69 | 78.38 | 80.68 | 79.56 | 78.95 | 77.88 | 82.09 |
| SaliencyMix | 0.2 | 79.75 | 79.64 | 79.12 | 77.66 | 80.72 | 78.63 | 78.77 | 77.51 | 84.35 |
| AttentiveMix+ | 2 | 79.62 | 80.14 | 78.91 | 78.41 | 81.69 | 81.53 | 80.54 | 79.60 | 84.34 |
| FMix | 0.2 | 78.91 | 79.91 | 79.69 | 79.50 | 79.87 | 78.99 | 79.02 | 78.24 | 84.21 |
| GridMix | 0.2 | 78.23 | 78.60 | 78.72 | 77.58 | 81.11 | 79.80 | 78.90 | 76.11 | 84.24 |
| ResizeMix | 1 | 79.56 | 79.19 | 80.01 | 79.23 | 79.56 | 79.78 | 80.35 | 79.73 | 84.87 |
| PuzzleMix | 1 | 79.96 | 80.82 | 81.13 | 81.10 | 81.69 | 82.84 | 82.85 | 82.93 | 85.02 |
| Co-Mixup† | 2 | 80.01 | 80.87 | 81.17 | 81.18 | 81.73 | 82.88 | 82.91 | 82.97 | 85.05 |
| AutoMix | 2 | 80.12 | 81.78 | 82.04 | 81.95 | 82.84 | 83.32 | 83.64 | 83.80 | 85.18 |
| SAMix | 2 | 81.21 | **81.97** | 82.30 | **82.41** | 83.81 | **84.27** | **84.42** | **84.31** | **85.50** |
| AdAutoMix | 1 | **81.55** | **81.97** | **82.32** | - | **84.40** | 84.05 | 84.42 | - | 85.32 |

Table A7: Top-1 accuracy (%), GPU memory (G), and total training time (h) of 600 epochs on CIFAR-100 training 200 and 600 epochs based on DeiT-S, Swin-T, and ConvNeXt-T with the DeiT training setting. Notice that all methods are trained on a single A100 GPU to collect training times and GPU memory.

| Methods | α | DeiT-Small | | | | Swin-Tiny | | | | ConvNeXt-Tiny | | | |
| | | 200 ep | 600 ep | Mem. | Time | 200 ep | 600 ep | Mem. | Time | 200 ep | 600 ep | Mem. | Time |
|---|---|---|---|---|---|---|---|---|---|---|---|---|---|
| Vanilla | - | 65.81 | 68.50 | 8.1 | 27 | 78.41 | 81.29 | 11.4 | 36 | 78.70 | 80.65 | 4.2 | 10 |
| Mixup | 0.8 | 69.98 | 76.35 | 8.2 | 27 | 76.78 | 83.67 | 11.4 | 36 | 81.13 | 83.08 | 4.2 | 10 |
| CutMix | 2 | 74.12 | 79.54 | 8.2 | 27 | 80.64 | 83.38 | 11.4 | 36 | 82.46 | 83.20 | 4.2 | 10 |
| DeiT | 0.8, 1 | 75.92 | 79.38 | 8.2 | 27 | 81.25 | 84.41 | 11.4 | 36 | 83.09 | 84.12 | 4.2 | 10 |
| ManifoldMix | 2 | - | - | 8.2 | 27 | - | - | 11.4 | 36 | 82.06 | 83.94 | 4.2 | 10 |
| SmoothMix | 0.2 | 67.54 | 80.25 | 8.2 | 27 | 66.69 | 81.18 | 11.4 | 36 | 78.87 | 81.31 | 4.2 | 10 |
| SaliencyMix | 0.2 | 69.78 | 76.60 | 8.2 | 27 | 80.40 | 82.58 | 11.4 | 36 | 82.82 | 83.03 | 4.2 | 10 |
| AttentiveMix+ | 2 | 75.98 | 80.33 | 8.3 | 35 | 81.13 | 83.69 | 11.5 | 43 | 82.59 | 83.04 | 4.3 | 14 |
| FMix | 1 | 70.41 | 74.31 | 8.2 | 27 | 80.72 | 82.82 | 11.4 | 36 | 81.79 | 82.29 | 4.2 | 10 |
| GridMix | 1 | 68.86 | 74.96 | 8.2 | 27 | 78.54 | 80.79 | 11.4 | 36 | 79.53 | 79.66 | 4.2 | 10 |
| ResizeMix | 1 | 68.45 | 71.95 | 8.2 | 27 | 80.16 | 82.36 | 11.4 | 36 | 82.53 | 82.91 | 4.2 | 10 |
| PuzzleMix | 2 | 73.60 | 81.01 | 8.3 | 35 | 80.33 | 84.74 | 11.5 | 45 | 82.29 | 84.17 | 4.3 | 53 |
| AlignMix | 1 | - | - | - | - | 78.91 | 83.34 | 12.6 | 39 | 80.88 | 83.03 | 4.2 | 13 |
| AutoMix | 2 | 76.24 | 80.91 | 18.2 | 59 | 82.67 | 84.05 | 29.2 | 75 | 83.30 | 84.79 | 10.2 | 56 |
| SAMix | 2 | **77.94** | **82.49** | 21.3 | 58 | **82.70** | **84.74** | 29.3 | 75 | **83.56** | **84.98** | 10.3 | 57 |
| TransMix | 0.8, 1 | 76.17 | 79.33 | 8.4 | 28 | 81.33 | 84.45 | 11.5 | 37 | - | - | - | - |
| SMMix | 0.8, 1 | 74.49 | 80.05 | 8.4 | 28 | 81.55 | - | 11.5 | 37 | - | - | - | - |
| Decoupled (DeiT) | 0.8, 1 | 76.75 | 79.78 | 8.2 | 27 | 81.10 | 84.59 | 11.4 | 36 | 83.44 | 84.49 | 4.2 | 10 |

Table A8: More evaluation metric (robustness and calibration) on CIFAR-100 with 200-epoch training, reporting top-1 accuracy (%)↑ (clean data, corruption data, and FGSM attacks) and calibration ECE (%)↓.

| Methods | $\alpha$ | DeiT-Small | | | | Swin-Tiny | | | |
|---|---|---|---|---|---|---|---|---|---|
| | | Clean | Corruption | FGSM | ECE↓ | Clean | Corruption | FGSM | ECE↓ |
| Vanilla | - | 65.81 | 49.31 | 20.58 | 9.48 | 78.41 | 58.20 | 12.87 | 11.67 |
| Mixup | 0.8 | 69.98 | 55.85 | 17.65 | 7.38 | 76.78 | 59.11 | 15.03 | 13.89 |
| CutMix | 2 | 74.12 | 55.08 | 12.53 | 6.18 | 80.64 | 57.73 | 18.38 | 10.95 |
| DeiT | 0.8, 1 | 75.92 | 57.36 | 18.55 | 5.38 | 81.25 | 62.21 | 15.66 | 15.68 |
| SmoothMix | 0.2 | 67.54 | 52.42 | 15.07 | 30.59 | 66.69 | 49.69 | 9.79 | 27.10 |
| SaliencyMix | 0.2 | 69.78 | 51.14 | 17.31 | 5.45 | 80.40 | 58.43 | 15.29 | 10.49 |
| AttentiveMix+ | 2 | 75.98 | 57.57 | 13.90 | 9.89 | 81.13 | 58.07 | 15.43 | 9.60 |
| FMix | 1 | 70.41 | 51.94 | 12.20 | 4.14 | 80.72 | 58.44 | 13.97 | 9.19 |
| GridMix | 1 | 68.86 | 51.11 | 8.43 | 4.09 | 78.54 | 57.78 | 11.07 | 9.37 |
| ResizeMix | 1 | 68.45 | 50.87 | 20.03 | 7.64 | 80.16 | 57.37 | 13.64 | 7.68 |
| PuzzleMix | 2 | 73.60 | 57.67 | 17.44 | 9.45 | 80.33 | 60.67 | 12.96 | 16.23 |
| AlignMix | 1 | - | - | - | - | 78.91 | 61.61 | 17.20 | **1.92** |
| AutoMix | 2 | 76.24 | 60.08 | 27.35 | 4.69 | 82.67 | **64.10** | 23.62 | 9.19 |
| SAMix | 2 | **77.94** | **61.91** | **30.35** | 4.01 | **82.70** | 62.19 | **23.66** | 7.85 |
| TransMix | 0.8, 1 | 76.17 | 59.89 | 22.48 | 8.28 | 81.33 | 62.53 | 18.90 | 16.47 |
| SMMix | 0.8, 1 | 74.49 | 59.96 | 22.85 | 8.34 | 81.55 | 62.86 | 19.14 | 16.81 |
| Decoupled (DeiT) | 0.8, 1 | 76.75 | 59.89 | 22.48 | 8.28 | 81.10 | 62.25 | 16.54 | 16.16 |

Table A9: Top-1 accuracy (%) on Tiny based on ResNet (R) and ResNeXt-32x4d (RX). Notice that † denotes reproducing results with the official implementation, while other results are implemented with OpenMixup.

| Backbones | $\alpha$ | R-18 | RX-50 |
|---|---|---|---|
| Vanilla | - | 61.68 | 65.04 |
| MixUp | 1 | 63.86 | 66.36 |
| CutMix | 1 | 65.53 | 66.47 |
| ManifoldMix | 0.2 | 64.15 | 67.30 |
| SmoothMix | 0.2 | 66.65 | 69.65 |
| AttentiveMix+ | 2 | 64.85 | 67.42 |
| SaliencyMix | 1 | 64.60 | 66.55 |
| FMix | 1 | 63.47 | 65.08 |
| GridMix | 0.2 | 65.14 | 66.53 |
| ResizeMix | 1 | 63.74 | 65.87 |
| PuzzleMix | 1 | 65.81 | 67.83 |
| Co-Mixup† | 2 | 65.92 | 68.02 |
| AutoMix | 2 | 67.33 | 70.72 |
| SAMix | 2 | 68.89 | 72.18 |
| AdAutoMix | 1 | **69.19** | **72.89** |

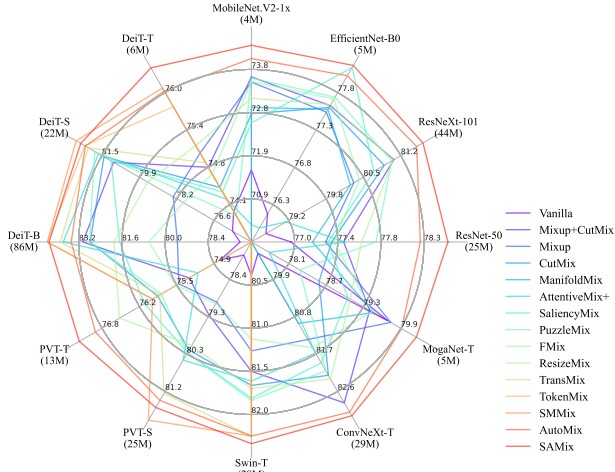

Figure A2: Radar plots of the top-1 accuracy of all evaluated mixup augmentation methods based on a variety of popular vision backbones on ImageNet-1K.

and optimize the models with a basic learning rate of 0.2 for 400 epochs with Cosine Scheduler. As shown in Table A9, all compared methods adopt ResNet-18 and ResNeXt-50 (32x4d) architectures training 400 epochs from the stretch on Tiny-ImageNet.

### B.3 DOWNSTREAM CLASSIFICATION BENCHMARKS

We further provide benchmarks on three downstream classification scenarios in 224×224 resolutions with ResNet architectures, as shown in Table A10.

**Benchmarks on Fine-grained Scenarios.** As for fine-grained scenarios, each class usually has limited samples and is only distinguishable in some particular regions. We conduct (a) transfer learning on CUB-200 and FGVC-Aircraft and (b) fine-grained classification with training from scratch on iNat2017 and iNat2018. For (a), we use transfer learning settings on fine-grained datasets, using PyTorch official pre-trained models as initialization and training 200 epochs by SGD optimizer with the initial learning rate of 0.001, the weight decay of 0.0005, the batch size of 16, the same data augmentation as ImageNet-1K settings. For (b) and (c), we follow Pytorch-style ImageNet-1K settings mentioned above, training 100 epochs from the stretch.

Table A10: Top-1 accuracy (%) of mixup image classification with ResNet (R) and ResNeXt (RX) variants on fine-grained datasets (CUB-200, FGVC-Aircraft, iNat2017/2018) and Places205.

| Method | Beta $\alpha$ | CUB-200 R-18 | CUB-200 RX-50 | FGVC-Aircraft R-18 | FGVC-Aircraft RX-50 | Beta $\alpha$ | iNat2017 R-50 | iNat2017 RX-101 | iNat2018 R-50 | iNat2018 RX-101 | Beta $\alpha$ | Places205 R-18 | Places205 R-50 |
|---|---|---|---|---|---|---|---|---|---|---|---|---|---|
| Vanilla | - | 77.68 | 83.01 | 80.23 | 85.10 | - | 60.23 | 63.70 | 62.53 | 66.94 | - | 59.63 | 63.10 |
| MixUp | 0.2 | 78.39 | 84.58 | 79.52 | 85.18 | 0.2 | 61.22 | 66.27 | 62.69 | 67.56 | 0.2 | 59.33 | 63.01 |
| CutMix | 1 | 78.40 | 85.68 | 78.84 | 84.55 | 1 | 62.34 | 67.59 | 63.91 | 69.75 | 0.2 | 59.21 | 63.75 |
| ManifoldMix | 0.5 | 79.76 | 86.38 | 80.68 | 86.60 | 0.2 | 61.47 | 66.08 | 63.46 | 69.30 | 0.2 | 59.46 | 63.23 |
| SaliencyMix | 0.2 | 77.95 | 83.29 | 80.02 | 84.31 | 1 | 62.51 | 67.20 | 64.27 | 70.01 | 0.2 | 59.50 | 63.33 |
| FMix | 0.2 | 77.28 | 84.06 | 79.36 | 86.23 | 1 | 61.90 | 66.64 | 63.71 | 69.46 | 0.2 | 59.51 | 63.63 |
| ResizeMix | 1 | 78.50 | 84.77 | 78.10 | 84.0 | 1 | 62.29 | 66.82 | 64.12 | 69.30 | 1 | 59.66 | 63.88 |
| PuzzleMix | 1 | 78.63 | 84.51 | 80.76 | 86.23 | 1 | 62.66 | 67.72 | 64.36 | 70.12 | 1 | 59.62 | 63.91 |
| AutoMix | 2 | 79.87 | 86.56 | 81.37 | 86.72 | 2 | 63.08 | 68.03 | 64.73 | 70.49 | 2 | 59.74 | 64.06 |
| SAMix | 2 | **81.11** | **86.83** | **82.15** | **86.80** | 2 | **63.32** | **68.26** | **64.84** | **70.54** | 2 | **59.86** | **64.27** |

**Benchmarks on Scenis Scenarios.** As for scenic classification tasks, we study whether mixup augmentations help models distinguish the backgrounds, which are less important than the foreground objects in commonly used datasets. We employ the PyTorch-style training setting like ImageNet-1K on Places205 (Zhou et al., 2014), optimizing models for 100 epochs with SGD optimizer, a basic learning rate of 0.1 with 256 batch size.

**Visualization of Training Stabiltities.** To further analyze training stability and convergence speed, we provided more visualization of the training epoch vs. top-1 validation accuracy of various Mixup augmentations on different datasets to support the conclusion of training convergence, as shown in Figure A3. These visualization results could be easily obtained by our analysis tools under `tools/analysis_tools`.

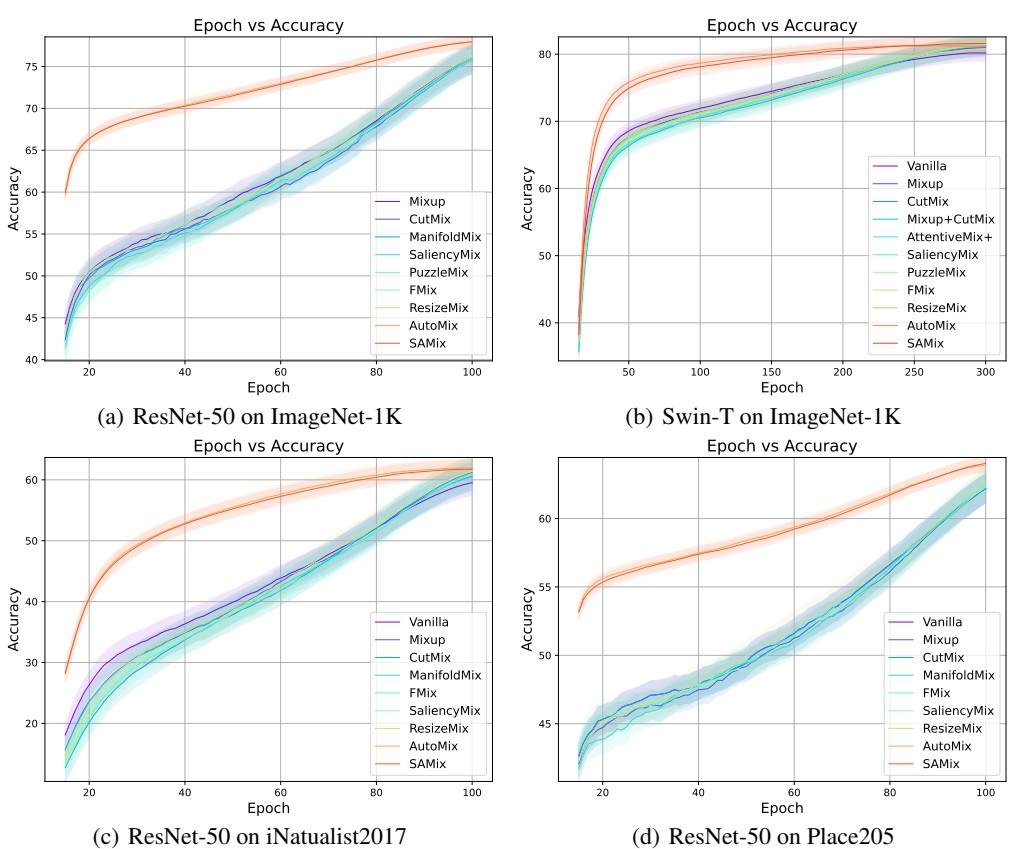

Figure A3: Training epoch *vs.* top-1 accuracy plots of various mixup methods on (a)(b) ImageNet-1K, (c) iNatualist2017, and (d) Place205 to further study training stability and convergence speed.

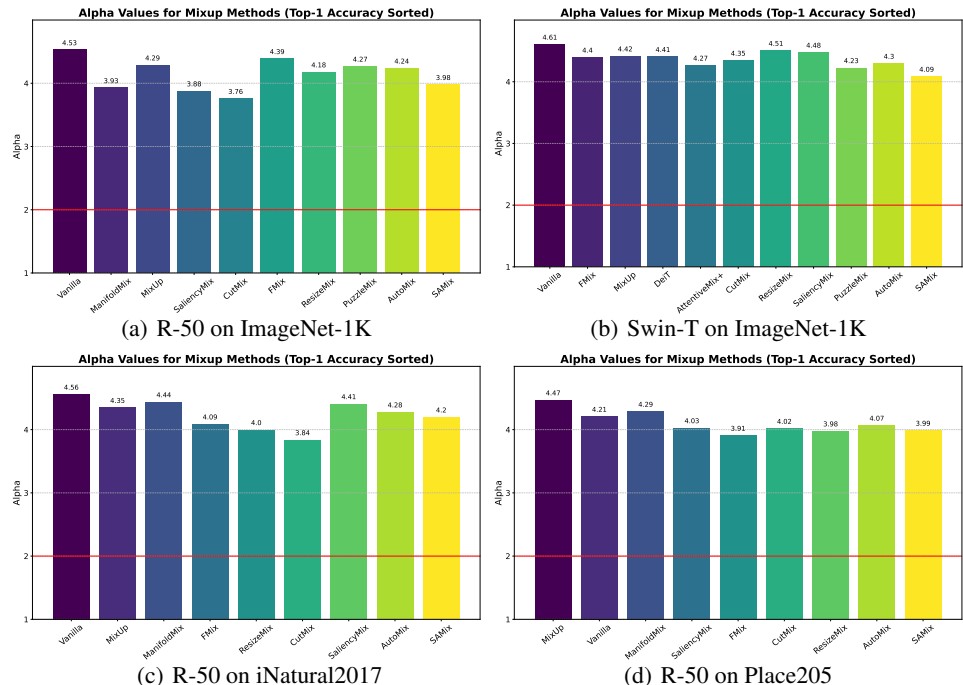

Figure A4: Explaination of learned ResNet-50 or Swin-T by various mixup methods with alpha metrics computed by `WeightWather` on (a)(b) ImageNet-1K, and (c) iNaturalist2017, and (d) Place205. In each figure, the bars are sorted with the top-1 accuracy from left to right. Empirically, the alpha metric indicates the degree of how well models fit the task, where alpha less than 2 or greater than 6 indicates the risk of overfitting and underfitting. (a)(b) On ImageNet-1K, favorable mixup methods (*e.g., dynamic* ones like AutoMix variants) prevent ResNet-50 (already had inductive bias) from overfitting while helping Swin-T learning better representations. (c) Since iNaturalist2017 is a smaller dataset with more difficult classes than ImageNet-1K, *dynamic* mixup methods tend to prevent overfitting to get better fine-grained classification performances. (d) Place205 with difficult scenic images, is two times larger than ImageNet-1K with iconic images. Therefore, it is likely to require mixup augmentations to encourage better fitting to scenic classification.

### B.4 TRANSFER LEARNING

**Object Detection.** We conduct transfer learning experiments with pre-trained ResNet-50 (He et al., 2016) and PVT-S (Wang et al., 2021) using mixup augmentations to object detection on COCO-2017 (Lin et al., 2014) dataset, which evaluate the generalization abilities of different mixup approaches. We first fine-tune Faster RCNN (Ren et al., 2015) with ResNet-50-C4 using Detectron2 (Wu et al., 2019) in Table A11, which is trained by SGD optimizer and multi-step scheduler for 24 epochs ($2\times$). The *dynamic* mixup methods (*e.g.,* AutoMix) usually achieve both competitive performances in classification and object detection tasks. Then, we fine-tune Mask R-CNN (He et al., 2017) by AdamW optimizer for 24 epochs using MMDetection (Chen et al., 2019) in Table A12. We have integrated Detectron2 and MMDetection into OpenMixup, and the users can perform the transferring experiments with pre-trained models and config files. Compared to *dynamic* sample mixing methods, recently-proposed label mixing policies (*e.g.,* TokenMix and SMMix) yield better performances with less extra training overheads.

**Semantic Segmentation.** We also perform transfer learning to semantic segmentation on ADE20K (Zhou et al., 2018) with Semantic FPN (Kirillov et al., 2019) to evaluate the generalization abilities to fine-grained prediction tasks. Following PVT (Wang et al., 2021), we fine-tuned Semantic FPN for 80K interactions by AdamW (Loshchilov & Hutter, 2019) optimizer with the learning rate of $2 \times 10^{-4}$ and a batch size of 16 on $512^2$ resolutions using MMSegmentation (Contributors, 2020b). Table A12 shows the results of transfer experiments based on PVT-S.

**Weakly Supervised Object Localization.** To study the localization ability of trained models more precisely, we follow CutMix (Yun et al., 2019) to evaluate the weakly supervised object localization

Table A11: Trasfer learning of object detection with ImageNet-1k pre-trained ResNet-50 backbone on COCO dataset.

| Method | IN-1K Acc | COCO mAP | COCO $AP^{bb}_{50}$ | COCO $AP^{bb}_{75}$ |
|---|---|---|---|---|
| Vanilla | 76.8 | 38.1 | 59.1 | 41.8 |
| Mixup | 77.1 | 37.9 | 59.0 | 41.7 |
| CutMix | 77.2 | 38.2 | 59.3 | 42.0 |
| ResizeMix | 77.4 | 38.4 | 59.4 | 42.1 |
| PuzzleMix | 77.5 | 38.3 | 59.3 | 42.1 |
| AutoMix | 77.9 | 38.6 | 59.5 | **42.2** |
| SAMix | **78.1** | **38.7** | **59.6** | 42.2 |

Table A12: Trasfer learning of object detection with Mask R-CNN and semantic segmentation with Semantic FPN with pre-trained PVT-S on COCO and ADE20K, respectively.

| Method | IN-1K Acc | COCO mAP | COCO $AP^{bb}_{50}$ | COCO $AP^{bb}_{75}$ | ADE20K mIoU |
|---|---|---|---|---|---|
| MixUp+CutMix | 79.8 | 40.4 | 62.9 | 43.8 | 41.9 |
| AutoMix | 80.7 | 40.9 | 63.9 | 44.1 | 42.5 |
| TransMix | 80.5 | 40.9 | 63.8 | 44.0 | 42.6 |
| TokenMix | 80.6 | **41.0** | **64.0** | 44.3 | **42.7** |
| TokenMixup | 80.5 | 40.7 | 63.6 | 43.9 | 42.5 |
| SMMix | **81.0** | **41.0** | 63.9 | **44.4** | **43.0** |

(WSOL) task on CUB-200 (Wah et al., 2011). The model localizes objects of interest based on the activation maps of CAM (Selvaraju et al., 2019) without bounding box supervision and calculates the maximal box accuracy with a threshold $\delta \in \{0.3, 0.5, 0.7\}$ as MaxBoxAccV2 (Choe et al., 2020). We provided the benchmarked results on CUB-200 in Table A13, where we found similar conclusions as the visualization of Grad-CAM in Sec. 4.2.

Table A13: MaxBoxAcc (%)↑ for the Weakly Supervised Object Localization (WSOL) task on CUB-200 based on ResNet architectures. Following CutMix (Yun et al., 2019), the model localizes objects of interest based on the activation maps of CAM (Selvaraju et al., 2019) without bounding box supervision and calculates the maximal box accuracy with a threshold $\delta \in \{0.3, 0.5, 0.7\}$ as MaxBoxAccV2 (Choe et al., 2020).

| Backbone | Vanilla | Mixup | CutMix | ManifoldMix | SaliencyMix | FMix | PuzzleMix | Co-Mixup | AutoMix | SAMix |
|---|---|---|---|---|---|---|---|---|---|---|
| R-18 | 49.91 | 48.62 | 51.85 | 48.49 | 52.07 | 50.30 | 53.95 | 54.13 | 54.46 | **57.08** |
| RX-50 | 53.38 | 50.27 | 57.16 | 49.73 | 58.21 | 59.80 | 59.34 | 59.76 | **61.05** | 60.94 |

## B.5 RULES FOR COUNTING THE MIXUP RANKINGS

We have summarized and analyzed a great number of mixup benchmarking results to compare and rank all the included mixup methods in terms of *performance*, *applicability*, and the *overall* capacity. Specifically, regarding the *performance*, we averaged the accuracy rankings of all mixup algorithms for each downstream task and averaged their robustness and calibration results rankings separately. Finally, these ranking results are averaged again to produce a comprehensive range of performance ranking results. As for the *applicability*, we adopt a similar ranking computation scheme considering the *time usage* and the *generalizability* of the methods. With the *overall* capacity ranking, we combined the performance and applicability rankings with a 1:1 weighting to obtain the final take-home rankings. For equivalent results, we take a tied ranking approach. For instance, if three methods are tied for first place, then the method that results in fourth place is recorded as second place by default. Finally, we provide the comprehensive rankings as shown in Table 1 and Table 5.

## C    REPRODUCTION COMPARISON

We provided the reproduction analysis of various mixup methods. Note that AutoMix (Qin et al., 2024), SAMix (Li et al., 2021), AdAutoMix (Qin et al., 2024), and Decouple Mix (Liu et al., 2022c) are **originally implemented in `OpenMixup`**, while the other popular mixup algorithms are reproduced based on their official source codes or descriptions. As shown in Table A14 and Table A15, the reproduced results are usually better than the original implementations because of the following reasons: To ensure a fair comparison, we follow the standard training settings for various datasets. Without changing the training receipts, we applied the effective implementations of the basic training components. For example, we employ a better implementation of the cosine annealing learning rate scheduler (updating by iterations) instead of the basic version (updating by epochs). On CIFAR-100, we utilize the `RandomCrop` augmentation with a "reflect" padding instead of the "zero" padding. On Tiny-ImageNet, we utilize `RandomResizedCrop` with the cropping ratio of 0.2 instead of `RandomCrop` in some implementations. On ImageNet-1K, we found that our reproduced results closely align with the reported performances, with any minor discrepancies (around $\pm 0.5\%$) attributable to factors such as random initialization and specific hardware configurations.

Table A14: Comparison of benchmark results reproduced by `OpenMixup` and the official implementations on CIFAR-100 and Tiny-ImageNet. We report the top-1 accuracy and the training epoch. Note that AutoMix (Qin et al., 2024), SAMix (Li et al., 2021), AdAutoMix (Qin et al., 2024), and Decouple Mix (Liu et al., 2022c) are **originally implemented in `OpenMixup`**. The reproduced results are usually better than the original implementations because we applied the effective implementations of the standard training settings without changing the training receipts.

| Method | Publication | CIFAR-100 (R18) | | Tiny-ImageNet (R18) | |
|---|---|---|---|---|---|
| | | **Ours** | Official | **Ours** | Official |
| MixUp(Zhang et al., 2018) | ICLR'2018 | **79.24 (1200)** | 76.84 (1200) | **63.86 (400)** | 58.96 (400) |
| CutMix(Yun et al., 2019) | ICCV'2019 | **78.29 (1200)** | 76.95 (1200) | **65.53 (400)** | 59.89 (400) |
| SmoothMix(ha Lee et al., 2020) | CVPRW'2020 | **78.69 (800)** | 78.14 (800) | **66.65 (400)** | - |
| GridMix(Baek et al., 2021) | PR'2020 | **78.72 (800)** | 78.09 (800) | **64.79 (400)** | 62.22 (400) |
| ResizeMix(Qin et al., 2023) | CVMJ'2023 | **79.19 (400)** | 79.05 (400) | **63.47 (400)** | 63.23 (400) |
| ManifoldMix(Verma et al., 2019) | ICML'2019 | **80.21 (1200)** | 79.98 (1200) | **64.15 (400)** | 60.24 (400) |
| FMix(Harris et al., 2020) | arXiv'2020 | **79.91 (400)** | 79.85 (400) | **63.47 (400)** | 61.43 (400) |
| AttentiveMix(Walawalkar et al., 2020) | ICASSP'2020 | **79.62 (200)** | 77.16 (200) | **64.01 (400)** | - |
| SaliencyMix(Uddin et al., 2020) | ICLR'2021 | **79.75 (200)** | 76.11 (200) | **64.60 (400)** | - |
| PuzzleMix(Kim et al., 2020) | ICML'2020 | **81.13 (800)** | 80.99 (800) | **65.81 (400)** | 63.48 (400) |
| AlignMixup(Venkataramanan et al., 2022) | CVPR'2022 | **82.27 (800)** | 82.12 (800) | **66.91 (400)** | 66.87 (400) |

Table A15: Comparison of reproduced results with `OpenMixup` and the official implementations on ImageNet-1K. We report the top-1 accuracy and the training epoch. Our reproduced results closely align with the reported performances, with any minor discrepancies (around $\pm 0.5\%$) attributable to factors such as random initialization and specific hardware configurations.

| Method | Publication | ImageNet-1K | | |
|---|---|---|---|---|
| | | Backbone | **Ours** | Official |
| MixUp (Zhang et al., 2018) | ICLR'2018 | R50 | **77.12 (100)** | 77.01 (100) |
| CutMix (Yun et al., 2019) | ICCV'2019 | R50 | **77.17 (100)** | 77.08 (100) |
| SmoothMix (ha Lee et al., 2020) | CVPRW'2020 | R50 | **77.84 (300)** | 77.66 (300) |
| GridMix (Baek et al., 2021) | PR'2020 | R50 | **78.50 (300)** | 78.25 (300) |
| ResizeMix (Qin et al., 2023) | CVMJ'2023 | R50 | **78.91 (300)** | 78.90 (300) |
| ManifoldMix (Verma et al., 2019) | ICML'2019 | R50 | **77.01 (100)** | 76.85 (100) |
| FMix (Harris et al., 2020) | arXiv'2020 | R50 | **77.19 (100)** | 77.03 (100) |
| AttentiveMix (Walawalkar et al., 2020) | ICASSP'2020 | DeiT-S | **80.32 (300)** | 77.50 (300) |
| SaliencyMix (Uddin et al., 2020) | ICLR'2021 | R50 | 78.46 (300) | **78.76 (300)** |
| PuzzleMix (Kim et al., 2020) | ICML'2020 | R50 | **77.54 (100)** | 77.51 (100) |
| AlignMixup (Venkataramanan et al., 2022) | CVPR'2022 | R50 | 79.32 (300) | **79.50 (300)** |
| TransMix (Chen et al., 2022) | CVPR'2022 | DeiT-S | **80.80 (300)** | 80.70 (300) |
| SMMix (Chen et al., 2023) | ICCV'2023 | DeiT-S | **81.10 (300)** | **81.10 (300)** |

