# OpenReview forum: "OpenMixup: Open Mixup Toolbox and Benchmark for Visual Representation Learning"
_ICLR.cc/2025/Conference — ICLR 2025 Conference Withdrawn Submission_

### Official Review · Reviewer_jWZD · 2024-10-28

**Soundness:** 2
**Presentation:** 2
**Contribution:** 2
**Rating:** 3
**Confidence:** 4

**Summary:**

The paper studies popular mixup strategies across different datasets and settings and finds that dynamic mixup strategies like SAMix are better across the spectrum than static ones like the original mixup. They also introduce a Python framework named OpenMixUp to promote full reproducibility for their findings and also to allow the community to easily experiment with mixup strategies.

**Strengths:**

1. Extensive pool of mixup strategies, including both static and dynamic ones (18 in total).
2. Going beyond standard classification benchmarks by including experiments on transfer learning (object detection, semantic segmentation).
3. An open discussion of their philosophy for implementing the OpenMixUp project, detailing the technical aspects.

**Weaknesses:**

1. Small datasets, not representative of the scale provided by current open datasets.
2. Architectural ablations are not very comprehensive to establish trends.
3. Many pieces of important information missing from the main text (such as results on non-classification benchmarks).

**Questions:**

1. I don't find strong motivation to do MixUp any longer when the community has already started moving away from self-supervised learning and towards (perhaps more scalable) approaches such as semi-supervised and weakly-supervised learning. I think this is especially true in the emergence of vision-language models across all kinds of scaling regimes. Maybe the authors could make a more compelling argument in the abstract or the introduction, discussing potential applications or benefits of mixup that are still valuable in current research directions.

2. The importance of MixUp isn't that evident in the area of vision-language models. Even when we're in the visual representation learning territory, with the emergence of models like Cambrian [1], the multimodal aspects of this paradigm cannot be neglected. In this paradigm, there hasn't been any benefit of using MixUp.

3. The datasets (such as COYO [2], MetaCLIP [3]) used in the study are not very representative of the scale current open datasets provide. It would have been better to repurpose at least one large-scale dataset (even if trimmed down) to study if the observations transfer well.

4. Most modern backbones and methods (ConvNeXt-v2 [4], HIERA [5], for example) employ MixUp during the fine-tuning stage. So, this could have been studied in this paper, i.e., take a pre-trained backbone and evaluate the mixup strategies.

5. Authors studied different transfer learning scenarios in their setup, including semantic segmentation and object detection. This could be made clearer in the main text of the paper because, from the main text, it's hard to know if the authors go beyond non-classification tasks.

6. L097 - L101 Per-step training latency is another important factor to consider. So, a better trade-off could be to include top-1 accuracy, memory consumption, and time. This gives a more complete picture of the gains/losses. I do see this being reflected in Figure 4. Perhaps the authors could make this more explicit to the readers, letting them know that this trade-off is available already.

7. L101 - L102 Do the static methods perform equally worse on other visual representation learning tasks? VTAB [6] is a representative benchmark that helps to evaluate the effectiveness of a method more completely. I think we need to consider this aspect here because it might so happen that a static method may perform poorly on one task but may perform better on the other. So, comparing the performance of different methods on an established benchmark (like VTAB) provides better signals.

8. MixUp is a training-only technique applied at the data-processing level. Why does OpenMixUp have to have a module for model architectures?

9. From Figure 4, it's not clear what defines the size of the circle markers.

10. When doing the ablations on architectures, I think a better way would have been to first group the architectures with respect to model families and then perform ablations. For example, ResNet [7], ResNeXT [8], could go to native CNN-based architectures. ConvNeXt [9] could go to the modern CNN family with transformer-alike components (like activations) and so on. In my opinion, this could have been followed in the presentation of the results.

References

[1] Cambrian-1: A Fully Open, Vision-Centric Exploration of Multimodal LLMs; Tong et al.; 2024.

[2] COYO-700M: Large-scale Image-Text Pair Dataset.

[3] Demystifying CLIP Data; Xu et al.; 2023.

[4] ConvNeXt V2: Co-designing and Scaling ConvNets with Masked Autoencoders; Woo et al.; 2023.

[5] Hiera: A Hierarchical Vision Transformer without the Bells-and-Whistles; Ryali et al.; 2023.

[6] A Large-scale Study of Representation Learning with the Visual Task Adaptation Benchmark; Zhai et al.; 2019.

[7] Deep Residual Learning for Image Recognition; He et al.; 2015.

[8] Aggregated Residual Transformations for Deep Neural Networks; Xie et al.; 2016.

---

### Official Review · Reviewer_24LQ · 2024-10-30

**Soundness:** 3
**Presentation:** 3
**Contribution:** 4
**Rating:** 5
**Confidence:** 4

**Summary:**

The paper introduces OpenMixup, a benchmark aimed at mixup augmentation in visual representation learning. OpenMixup evaluates 18 different mixup techniques on 11 image datasets and offers a modular, open-source codebase to streamline data pre-processing, mixup implementation, training, and evaluation. Through a range of experiments, the paper examines how each mixup method balances performance and computational demands, offering recommendations on when and how to use different mixup strategies. Already in use within the computer vision field, the codebase highlights areas for further exploration in mixup augmentation.

**Strengths:**

1. OpenMixup serves as a detailed benchmark for mixup augmentation, systematically testing 18 mixup methods across 11 diverse image datasets. This broad evaluation creates a solid foundation for comparing mixup strategies, supporting reproducibility, and enabling comprehensive comparisons within the field.
2. The codebase is modular, open-source, and available to the public, complete with standardized components for data pre-processing, mixup implementation, network selection, optimization, and evaluation. Useful for community.
3. Detailed insights into the performance and complexity trade-offs of different mixup methods, emphasizing the generalizability of dynamic mixup methods compared to static ones, their robustness against input corruptions and adversarial attacks, and their versatility across various neural network architectures.
4. Rankings of mixup methods are offered based on performance, applicability, and overall effectiveness, making it easier for researchers and practitioners.
5. Well organized and well written - makes it easy to understand.

**Weaknesses:**

1. Computational analysis of dynamic mixup methods as they perform better than static however detailed and concrete insights on computational overhead is missing which makes difficult to make decision on selection. These limitation should be mitigated with deep trade off analysis.
2. The paper could benefit from an objective evaluation of hyperparameter sensitivity in dynamic mixup methods. A clearer understanding of how performance varies with different hyperparameter settings would be valuable for practitioners in selecting methods suited to their specific requirements. I would like to get detailed comment on it from author as it will be practically useful for community.
3. Transferability to object detection and semantic segmentation tasks from ImageNet model is encouraging but not sufficient. Other computer vision task e.g. depth estimation should be considered and more generalized transferability pattern across task from pixel level to holistic, should be analyzed.

**Questions:**

I have described the questions with weakness clearly and would like to review the further response from authors. Upon receiving reasonable response with required information, I can change the score.

---

### Official Review · Reviewer_H7Cm · 2024-11-03

**Soundness:** 2
**Presentation:** 2
**Contribution:** 2
**Rating:** 3
**Confidence:** 4

**Summary:**

This paper provided a holistoic analysis study over a line of work in mixup and its variants. This work conduct 18 different mixup approaches across 11 datasets and open-source the codes for the community. Furthermore, this work comprehesively compares those 18 mixup approaches in terms of performance, efficiency, generalization, etc.

**Strengths:**

1. This is a very comprehensive study regarding the mixup and its variants, and this work provides a fair platform for the researchers to compare it.

2. As there are many mixup approaches, the authors provided an overall rating based on performance and applicability to guide the users to select proper methods given users' requirements.

**Weaknesses:**

1. This work conduct a comprehensive study; however, there is no surprising findings or new insight compared to those original papers; e.g., the online-mixup methods take more times and more memory but result in better performance in general, which is reasonable and known.

2. Given this paper would like to cover a lot of spectrum, the authors would like to mention all of the results they produced; then, the authors heavily use the appendix as a part of main text, e.g. at sec 4.2, many content is based on the table/figure in the appendix. For me, this means that the main paper is incomplete.

3. I am confused by the combination of dataset and model architecture, I understand that there could be many combinations but as an analysis paper, it is important to control the variables, e.g, in Table 3, the authors report the results on cifar100 with WRN-28-8, and never use WRN anymore; instead in figure 4, DeiT-S and ConNeXt-T are used. Another example, for Figure 5c, the authors report results for ResNet-50 on imagenet but mostly use DeiT-S in figure 4 and 6.

**Questions:**

See weaknesses

Questions:
1. at line 245, here says 17 mixup method and 12 datasets, it is inconsistent with the number mentioned in the abstract.

2. What is x-axis in Figure 5c?

3 . Why the overall ranking in Table 1 and 5 are inconsistent? e.g., DeiT.

---

### Official Review · Reviewer_VXep · 2024-11-03

**Soundness:** 3
**Presentation:** 3
**Contribution:** 3
**Rating:** 6
**Confidence:** 2

**Summary:**

This paper presents a study over mixup augmentations. 18 models are trained from scratch and evaluated in 12 vision classification datasets. Besides showing task results it also provides analysis such as practical recommendations, training curves and compute/memory usage and loss landscape. The codebase will be open-sourced and facilitate reproduction and further mixup investigation and comparisons.

**Strengths:**

- This paper is clearly written and provides a good background of existing mixup methods.

- It compares 18 different methods in 12 vision tasks trained from scratch in the same codebase. This is a significant and fair comparison with potential benefits to the community by bringing clear comparison between all the methods.

- The code will be available, allowing others to try ideas on top of it.

- It clearly passes comparison messages such as: dynamic policy performing better than static ones, a ranking of methods based on performance and how applicable they are (e.g. depend on specific architecture), compute and memory cost of the methods.

**Weaknesses:**

- The paper spends a significant amount of space describing the OpenMixup software, it is indeed described as a main contribution of the paper, but overall software description and how its structure to be reused by others seems better left for software documentation and not as main contribution. In particular no significant design decisions seem novel to the mixup problem that deserves attention. (Which is good, software should not be forced to be novel and unexpected). Instead of that I would have preferred to see further exploration/guidance of when to use mixup.

- Figure 4 is really hard to parse, sharing legend and using more space might make it better at conveying information.

- Although the information is there, I found it hard to parse when/whether there is an absolute significant gain on going from vanilla->static->dynamic, that and some other general conclusions when it is worth to use it would be useful information for a practitioner.

**Questions:**

- The comparisons in this work are when training from scratch, is mixup relevant for practitioners who initialise from pretrained networks trained on larger datasets than ImageNet-1k?

- How do mixup benefits behave when the number of training examples gets reduced/increased?

---

### Note · Authors · 2025-01-16

**Comment:**

We sincerely thank the reviewers and ACs of ICLR2025 for their efforts and detailed reviews. Although the research of mixup augmentations could be classical and ``old-fashioned" in the era of AIGC and LLM/MLLM, we believe our benchmarks and codebase could still benefit some deep learning applications of downstream tasks. Therefore, we decided to withdraw the manuscript and keep on updating OpenMixup.

Warm regards,

The Authors of OpenMixup

**Withdrawal Confirmation:**

I have read and agree with the venue's withdrawal policy on behalf of myself and my co-authors.